

**Evaluating model outputs using integrated global speleothem records of**
**climate change since the last glacial**
Laia Comas-Bru[1,2,*], Sandy P. Harrison[1], Martin Werner[3], Kira Rehfeld[4], Nick Scroxton[5], Cristina Veiga-
Pires[6] and SISAL working group members
Corresponding author: Laia Comas-Bru (l.comasbru@reading.ac.uk)
1- School of Archaeology, Geography & Environmental Sciences, Reading University, Whiteknights,
Reading, RG6 6AH, UK
2- UCD School of Earth Sciences. University College Dublin, Belfield. Dublin 4, Ireland.
3- Alfred Wegener Institute. Helmholtz Centre for Polar and Marine Research. Division Climate
Science - Paleoclimate Dynamics. Bussestr. 24, D-27570 Bremerhaven, Germany
4- Institute of Environmental Physics, Ruprecht-Karls-Universität Heidelberg, Im Neuenheimer Feld
229, 69120 Heidelberg, Germany
5- Department of Geosciences, University of Massachusetts Amherst, 611 North Pleasant Street,
01003-9297 Amherst, MA, USA
6- Universidade do Algarve Faculdade de Ciências do Mar e do Ambiente - FCMA Centro de
Investigação Marinha e Ambiental - CIMA Campus de Gambelas 8005-139 Faro Portugal





**Abstract:** Although quantitative isotopic data from speleothems has been used to evaluate isotope-
enabled model simulations, currently no consensus exists regarding the most appropriate
methodology through which achieve this. A number of modelling groups will be running isotope-
enabled palaeoclimate simulations in the framework of the Coupled Model Intercomparison Project
Phase 6, so it is timely to evaluate different approaches to use the speleothem data for data-model
comparisons. Here, we accomplish this using 456 globally-distributed speleothem $\delta^{18}O$ records from
an updated version of the Speleothem Isotopes Synthesis and Analysis (SISAL) database and
palaeoclimate simulations generated using the ECHAM5-wiso isotope-enabled atmospheric
circulation model. We show that the SISAL records reproduce the first-order spatial patterns of
isotopic variability in the modern day, strongly supporting the application of this dataset for evaluating
model-derived isotope variability into the past. However, the discontinuous nature of many
speleothem records complicates procuring large numbers of records if data-model comparisons are
made using the traditional approach of comparing anomalies between a control period and a given
palaeoclimate experiment. To circumvent this issue, we illustrate techniques through which the
absolute isotopic values during any time period could be used for model evaluation. Specifically, we
show that speleothem isotope records allow an assessment of a model's ability to simulate spatial
isotopic trends and the degree to which the model reproduces the observed environmental controls
of isotopic spatial variability. Our analyses provide a protocol for using speleothem isotopic data for
model evaluation, including screening the observations, the optimum period for the modern
observational baseline, and the selection of an appropriate time-window for creating means of the
isotope data for palaeo time slices.
**1. Introduction**
Earth System Models (ESMs) are routinely used to project the consequences of current and future
anthropogenic forcing of climate, and the impacts of these projected changes on environmental
services (e.g., Christensen et al., 2013;Collins et al., 2013;Kirtman et al., 2013;Field, 2014). ESMs are





routinely evaluated using modern and historical climate data. However, the range of climate variability
experienced during the period for which we have reliable historic climate observations is small, much
smaller than the amplitude of changes projected for the 21st century. Radically different climate states
in the geologic past provide an opportunity to test the performance of ESMs in response to very large
changes in forcing, changes that in some cases are as large as the expected change in forcing at the
end of the 21st century (Braconnot et al., 2012). The use of "out-of-sample" testing (Schmidt et al.,
2014) is now part of the evaluation procedure of the Coupled Model Intercomparison Project (CMIP).
Several palaeoclimate simulations are being run by the Palaeoclimate Modelling Intercomparison
Project (PMIP) as part of the sixth phase of CMIP (CMIP6-PMIP4), including simulations of the Last
Millennium (LM, 850–1850 CE, *past1000*), mid-Holocene (MH, ca. 6000 yrs BP, *midHolocene*) Last
Glacial Maximum (LGM, ca. 21,000 yrs BP, *lgm*), the Last Interglacial (LIG, ca. 127,000 yrs BP, *lig127k*)
and the mid-Pliocene Warm Period (mPWP, ca. 3.2 M yrs BP, *midPliocene-eoi400*) (Kageyama et al.,

54    2017).

Although these CMIP6-PMIP4 time periods were selected because they represent a range of different
climate states, the choice also reflects the fact that global syntheses of palaeoenvironmental and
palaeoclimate observations exist across them, thereby providing the opportunity model
benchmarking (Kageyama et al., 2017). However, both the geographic and temporal coverage of the
different types of data is uneven. Ice core records are confined to polar and high-altitude regions and
provide regionally to globally integrated signals of forcings and climatic responses. Marine records
provide a relatively comprehensive coverage of the ocean state for the LGM, but low rates of
sedimentation mean they are less informative about the more recent past (Hessler et al., 2014). Lake
records provide qualitative information of terrestrial hydroclimate, but the most comprehensive
source of quantitative climate information over the continents is based on statistical calibration of
pollen records (see e.g., Bartlein et al., 2011). However, pollen preservation requires the long-term
accumulation of sediments under anoxic conditions and is consequently limited in semi-arid, arid and
highly dynamic wet regions such as in the tropics.



Oxygen isotopic records ($\delta^{18}O$) from speleothems, secondary carbonate deposits that form in caves
from water that percolates through carbonate bedrock (Atkinson, 1977;Fairchild and Baker, 2012),
provide an alternative source of information about past terrestrial climates. Although there are
hydroclimatic limits on the growth of speleothems, their distribution is largely constrained by the
existence of suitable geological formations and they are found growing under a wide range of climate
conditions, from extremely cold climates in Siberia (Vaks et al., 2013) to arid regions of Australia
(Treble et al., 2017). Therefore, speleothems have the potential to provide information about past
terrestrial climates in regions for which we do not have (and are unlikely to have) information from
pollen. As is the case with pollen, where quantitative climate reconstructions must be obtained
through statistical or forward modelling approaches (Bartlein et al., 2011), the interpretation of
speleothem isotope records in terms of climate variables is in some cases not straightforward
(Fairchild and Baker, 2012;Lachniet, 2009). However, some ESMs now use water isotopes as tracers
for the diagnosis of hydroclimate (Werner et al., 2016;Tindall et al., 2009;Schmidt et al., 2007), and
this opens up the possibility of using speleothem isotopic measurements directly for comparison with
model outputs. At least six modelling groups are planning isotope-enabled palaeoclimate simulations
as part of CMIP6-PMIP4.
As with other model evaluation studies, much of the diagnosis of isotope-enabled ESMs has focused
on modern day conditions (e.g., Joussaume et al., 1984;Hoffmann et al., 1998;Noone and Simmonds,
2002;Schmidt et al., 2007;Roche, 2013;Xi, 2014;Risi et al., 2016;Hu et al., 2018;Jouzel et al.,
2000;Hoffmann et al., 2000). However, isotope-enabled models have also been used in a
palaeoclimate context (e.g., Schmidt et al., 2007;LeGrande and Schmidt, 2008;LeGrande and Schmidt,
2009;Caley and Roche, 2013;Caley et al., 2014;Jasechko et al., 2015;Werner et al., 2016;Langebroek
et al., 2011;Zhu et al., 2017). The evaluation of these simulations has often focused on isotope records
from polar ice cores and from marine environments. Where use has been made of speleothem
records, the comparison has generally been based on a relatively small number of the available
records. Furthermore, all of the comparisons make use of an empirically-derived correction for the



temperature-dependence fractionation of calcite $\delta^{18}O$ at the time of speleothem formation that is
based on synthetic carbonates (Kim and O'Neil, 1997). This fractionation is generally poorly
constrained (McDermott, 2004;Fairchild and Baker, 2012), does not account for any non-equilibrium
of kinetic fractionation at the time of deposition and is not suitable for aragonite samples. Thus, using
a single standard correction and not screening records for mineralogy introduces uncertainty into the
data-model comparisons.
SISAL (Speleothem Isotopes Synthesis and Analysis), an international working group under the
auspices of the Past Global Changes (PAGES) project (http://pastglobalchanges.org/ini/wg/sisal), is an
initiative to provide a reliable, well-documented and comprehensive synthesis of isotopic records
from speleothems worldwide (Comas-Bru and Harrison, 2019). The first version of the SISAL database
(SISALv1: Atsawawaranunt et al., 2018a;Atsawawaranunt et al., 2018b) included 381 speleothem-
based isotope records and metadata to facilitate quality control and record selection. A major
motivation for the SISAL database was to provide a tool for benchmarking of palaeoclimate
simulations using isotope-enabled models.
In this paper, we examine a number of issues that need to be addressed in order to use the SISAL data
for model evaluation in the palaeoclimate context and make recommendations about robust
approaches that should be used for model evaluation in CMIP6-PMIP4. We focus on the MH and LGM
time periods, partly because the *midHolocene* and *lgm* experiments are the "entry cards" for the
CMIP6-PMIP4 simulations and partly because these are the PMIP time periods with the best coverage
of speleothem records. We use an updated version of the SISAL database (SISALv1b: Atsawawaranunt
et al., 2019) and simulations made with the ECHAM5-wiso isotope-enabled atmospheric circulation
model (Werner et al., 2011) to explore the various issues in making data-model comparisons.
Section 2 introduces the data and the methods used in this study. Section 2.1 introduces the isotope-
enabled model simulations for the modern (1958–2013), the *midHolocene* and the *lgm* experiments,
explains the methods used to calculate weighted simulated $\delta^{18}O$ values, and provides information



about the construction of time-slices. Section 2.2 presents the modern observed $\delta^{18}O$ in precipitation
($\delta^{18}O_p$) used. Section 2.3 introduces the speleothem isotopic data from the SISAL database and
explains the rationale for screening records. Section 3 describes the results of the analyses, specifically
the spatio-temporal coverage of the SISAL records (Section 3.1), the representation of modern
conditions (Section 3.2), anomaly-mode time-slice comparisons (Section 3.3), and the comparison of
$\delta^{18}O$ gradients in absolute values along spatial transects to test whether the model accurately records
regional variations in $\delta^{18}O$ (Section 3.4). Section 4 provides a protocol for using speleothem isotopic
records for data-model comparisons and section 5 summarises our main conclusions.
**2.   Methods**

**2.1.  Model simulations**

ECHAM5-wiso (Werner et al., 2011) is the isotope-enabled version of the ECHAM5 atmosphere GCM
(Roeckner et al., 2003;Roeckner et al., 2006;Hagemann et al., 2006). The water cycle in ECHAM5
contains formulations for evapotranspiration of terrestrial water, evaporation of ocean water, and the
formation of large-scale and convective clouds. Vapour, liquid, and frozen water are transported
independently within the atmospheric advection scheme. The stable water isotope module in
ECHAM5 computes the isotopic signal of different water masses through the entire water cycle,
including in precipitation and soil water.
ECHAM5-wiso was run for 1958–2013, using an implicit nudging technique to constrain simulated
fields of surface pressure, temperature, divergence and vorticity to the corresponding ERA-40 and
ERA-Interim reanalysis fields (Butzin et al., 2014). The *midHolocene* simulation (Wackerbarth et al.,
2012) was forced by orbital parameters and greenhouse gas concentrations appropriate to 6 ka
following the PMIP3 protocol (https://pmip3.lsce.ipsl.fr). The control simulation has modern values
for the orbital parameters and greenhouse gas (GHG) concentrations (Wackerbarth et al., 2012). The
change in sea surface temperatures (SST) and sea ice cover between 6 ka and the pre-industrial period



were calculated from 50-year averages from each interval extracted from a transient Holocene
simulation performed with the fully coupled ocean-atmosphere Community Climate System Model
CCSM3 (Collins et al., 2006). The anomalies were then added to the observed modern SST and sea ice
cover data to force the *midHolocene* simulation (Wackerbarth et al., 2012). For the *lgm* experiment
(Werner et al., 2018), orbital parameters, GHG concentrations, land-sea distribution, and ice sheet
height and extent followed the PMIP3 guidelines. Climatological monthly sea ice coverage and SST
changes were prescribed from the GLAMAP dataset (Paul and Schäfer-Neth, 2003). A uniform glacial
enrichment of sea surface water and sea ice of +1‰ ($\delta^{18}$O) and +8‰ ($\delta$D) on top of the present-day
isotopic composition of surface seawater was applied. For the ocean surface sate of the corresponding
control simulation, monthly climatological SST and sea ice cover for the period 1979-1999 were
prescribed. All the ECHAM5-wiso simulations were run at T106 horizontal grid resolution (approx.
1.1°x1.1°) with 31 vertical levels. The *midHolocene* and *lgm* experiments were run for 12 and 22 years,
respectively. Model anomalies for the MH and the LGM were calculated as the differences between
the MH/LGM simulation and the corresponding control simulations. We also calculated the anomaly
between the LGM and MH (LGM-MH), taking account of the difference between their control
simulations. We constructed simulated isotope anomalies by averaging the last 10 (*midHolocene*) and
20 (*lgm*) years of the simulations.
At best, the speleothem isotopic signal will be an average of the precipitation $\delta^{18}$O ($\delta^{18}$O$_p$) signals
weighted towards those months when precipitation is greatest. However, the signal is transmitted via
the karst system, and is therefore modulated by storage in the soil, recharge rates, mixing in the
subsurface, and varying residence times - ranging from hours to years (e.g. Breitenbach et al.,
2015;Riechelmann et al., 2017). These factors could all exacerbate differences between observations
and simulations. We investigated whether weighting the simulated $\delta^{18}$O signals by soil moisture or
recharge amount provided a better comparison measure than weighting by precipitation amount by
calculating three indices: (i) $\delta^{18}$O$_p$ weighted according to monthly precipitation amount (w$\delta^{18}$O$_p$); (ii)
$\delta^{18}$O$_p$ weighted according to the potential recharge amount calculated as precipitation minus



evaporation (P-E) for months where P-E > 0 (w$\delta^{18}O_{recharge}$); and (iii) soil water $\delta^{18}O$ weighted according
to soil moisture amount (w$\delta^{18}O_{sw}$). To investigate the impact of transit time on the comparisons, we
smoothed the simulated w$\delta^{18}O$ using a range of smoothing from 1–20 years. Finally, we investigated
whether differences in elevation between the model grid and speleothem records had an influence
on the quality of the data-model comparisons by applying an elevational correction of -2.5‰/km
(Lachniet, 2009) to the simulated w$\delta^{18}O$.

**2.2. Modern observations**

We use two sources of modern isotope data for assessment purposes: (i) $\delta^{18}O_p$ measurements from
the Global Network of Isotopes in Precipitation (GNIP) database (IAEA/WMO, 2018) and (ii) a gridded
dataset of global water isotopes from the Online Isotopes in Precipitation Calculator (OIPC: Bowen,
2018;Bowen and Revenaugh, 2003).
The GNIP database provides raw monthly $\delta^{18}O_p$ values for some part of the interval 03/1960 to
08/2017 for 977 stations. Individual stations have data for different periods of time and there are gaps
in most individual records; only two stations have continuous data for over 50 years and both are in
Europe (Valentia Observatory, Ireland, and Vienna Hohe-Warte, Austria).  Most GNIP stations are
more than 0.5° away from the SISAL cave sites, precluding a direct global comparison between GNIP
and SISAL records. However, the GNIP data can be used to examine simulated interannual variability.
Annual $\delta^{18}O$ averages were calculated from GNIP stations with at least 10 months of data per year and
5 or more years of data. Annual $\delta^{18}O_p$ data was extracted from the ECHAM5-wiso simulations at the
location of the GNIP stations for the years for which GNIP data is available at each station. We exclude
GNIP stations from coastal locations that are not land in the ECHAM5-wiso simulation. This dual
screening results in only 450 of the 977 GNIP stations being used for comparisons. Boxplots are
calculated with the standard deviation of annual $\delta^{18}O_p$ data.



The OIPC dataset provides a gridded long-term global (1960–2017) record of modern $w\delta^{18}O_p$, based
on combining data from 348 GNIP stations covering part or all the period 1960–2014 (IAEA/WMO,
2017) and other $w\delta^{18}O_p$ records from the Water Isotopes Database (Waterisotopes Database, 2017).
The OIPC data can be used to evaluate spatial patterns in both the SISAL records and the simulations.
**2.3. Speleothem isotope data**
We use an updated SISAL database (SISALv1b: Atsawawaranunt et al., 2019), which provides revised
versions of 45 records from SISALv1 and includes 60 new records (Table 1). SISALv1b has isotope
records from 455 speleothems from 211 cave sites distributed worldwide. Because the isotopic
fractionation between water and $CaCO_3$ differs between calcite and aragonite,  we only use calcite
speleothems or aragonite speleothems where the correction to calcite values was made by the original
authors for simplicity. However, using the reformulated aragonite $\delta^{18}O$-water equation of Grossman
and Ku (1986) from Lachniet (2015) would allow the incorporation of the currently small number of
aragonite records from the SISAL database to the data-model comparison. As a result of this screening,
we use 370 speleothem records from 174 cave sites for comparisons. However, the number of
speleothem records covering specific periods (i.e., modern, MH, LGM) is considerably lower.
Recent data suggests that many calcite speleothems are precipitated out of isotopic equilibrium with
waters (Daëron et al., 2019). Therefore, we have converted SISAL data to its drip-water equivalent
using an empirical speleothem-based fractionation factor that accounts for any non-equilibrium of
kinetic fractionation that may arise in the precipitation of calcite speleothems in caves (Tremaine et
al., 2011). We use the V-PDB to V-SMOW conversion from Coplen et al. (1983) (as in Sharp, 2007).
$\delta^{18}O_{calcite\_SMOW} = 0.97002 \cdot \left(\delta^{18}O_{calcite\_PDB} + 29.98\right)$
$\delta^{18}O_{dripw\_SMOW} = \delta^{18}O_{calcite\_SMOW} - \left(\left(\frac{16.1 \cdot 1000}{T}\right) - 24.6\right)$     (T in K)





We have used mean annual surface air temperature from CRU-TS4.01 (Harris et al., 2014) for the OIPC
comparison and ECHAM5-wiso simulated mean annual temperature for the SISAL-model comparison
as a surrogate of modern and past cave air temperature (Moore and Sullivan, 1997) Uncertainties are
introduced in this conversion from several unknown factors such as cave temperature and $pCO_2$ of
soil.
We compare the modern temporal variability in the SISAL records with ECHAM5-wiso by extracting
simulated $w\delta^{18}O_p$ at the cave site location for all the years for which there are speleothem isotope
samples. The speleothem isotope ages were rounded to exact calendar years for this comparison.
Data-model comparisons are generally made by comparing anomalies between a control period and
a palaeoclimate simulation with data anomalies with respect to a modern baseline. There is no agreed
standard defining the interval used as a modern baseline for palaeoclimate reconstructions. Some
studies have used modern observational datasets which cover a specific and limited period of time
and some use the late 20th century as a reference. We investigate the appropriate choice of modern
baseline for the speleothem records by comparing the interval centred on 1850 CE with alternative
intervals covering the late 20th century, specifically 1961-1990 and 1850–1990 CE, and we assess the
impact of these choices on both mean $\delta^{18}O$ values and the number of records available for
comparison. The MH time slice was defined as 6,000 ±500 yrs BP (where present is 1950 CE) and the
LGM time slice as 21,000 ±1,000 yrs BP, following the conventional definitions of these intervals used
in the construction of other benchmark palaeoclimate datasets (e.g., MARGO project members,
2009;Bartlein et al., 2011). However, we also examined the impact of using shorter intervals for each
time slice. In addition to calculating LGM and MH anomalies with respect to modern, we also
calculated the anomaly between the LGM and MH (LGM-MH).
We use the published age-depth models for each speleothem record. There is no information about
the temporal uncertainties on individual isotope samples for most of the records in SISALv1b. This
precludes a general assessment of the impact of temporal uncertainties on data-model comparisons.



We assess these impacts for the LGM for two records (entity BT-2 from Botuverá cave: Cruz et al.,
2005;and entity SSC01 from Gunung-buda cave: Partin et al., 2007) for which new age-depth models
have been prepared using COPRA (Breitenbach et al., 2012). We created 1,000-member ensembles of
the age-depth relationship using the original author's choice of radiometric dates and *pchip*
interpolation. Isotope ratio means were calculated using time windows of increasing width (100 to
3,000 years) around 21 kyrs BP for the original age-depth model, the COPRA median age model, and
all ensemble members. All COPRA-based uncertainties have been projected to the chronological axes.
To explore the use of absolute isotopic data for model evaluation, we extracted absolute data for six
transects illustrating key features of MH and LGM geographic isotopic patterns. The MH transects run
from NW to SE across America, NW to SE across SE Asia, and N-S across southern Europe and northern
Africa. The LGM transects run N-S from central Europe to southern Africa, from NW to SE in America,
and N-S from China to northern Australia. Each transect follows the great circle line between two
locations. The longitudinal span of each transect varies to maximise the number of SISAL records
included. We extracted model outputs for the same transects, using the model land/sea mask to
remove ocean grid cells. The simulated absolute values were extracted along the great circle lines at
1.12° steps to match the model grid size. Comparisons are made between the SISAL mean $\delta^{18}O$ value
and the simulated $w\delta^{18}O_p$ values averaged within a longitudinal range. We also compare simulated
mean annual surface air temperature (MAT) and mean annual precipitation (MAP) with pollen-based
quantitative reconstructions of MAT and MAP from Bartlein et al. (2011). The pollen-based anomalies
have been converted to absolute values by adding the CRU-TS4.01 climatology (Harris et al., 2014).
Speleothem growth is inhibited in very dry climates, so the presence/absence of speleothems has
been interpreted as a direct indication of climate state (Gascoyne et al., 1983;Vaks et al., 2006;Vaks
et al., 2013). Speleothem distribution through time approximates an exponential curve in many
regions around the world (e.g., Ayliffe et al., 1998;Jo et al., 2014;Scroxton et al., 2016). This
relationship suggests that the natural attrition of stalagmites is independent of the age of the



specimens and approximately constant through time, despite potential complications from erosion,
climatic changes and sampling bias. The underlying exponential curve can, therefore, be thought of as
a prediction of the number of expected stalagmites given the existing population. Intervals when
climate conditions were more/less favourable to speleothem growth can then be identified from
changes in the population size by subtracting this underlying exponential curve (Scroxton et al., 2016).
We apply this approach at a global level to the unscreened SISAL data by counting the number of
individual caves with stalagmite growth during every 1,000-yr period from 500 kyrs BP to the present.
Growth was indicated by a stable isotope sample at any point in each 1,000-year bin, giving 3,866 data
points distributed in 500 bins. We use cave numbers, rather than the number of individual
speleothems, to minimise the risk of over-sampled caves influencing the results. Random resampling
(100,000) of the 3,866 data points was used to derive 95% and 5% confidence intervals. The number
of speleothems cannot be reliably predicted by a continuous distribution when numbers are low, so
we do not consider intervals prior to 266 kyrs BP – the most recent interval with less than four records.
**3.  Results**
**3.1.  Spatial and temporal coverage of speleothem records**
There are many regions of the world where the absence of carbonate lithologies means that there will
never be speleothem records (Fig. 1a). Nevertheless, SISALv1b represents a substantial improvement
in spatial coverage compared to SISALv1, particularly for Australasia and Central and North American
(Fig. 1a, Table 1), and the sampling for regions such as Europe and China is quite dense. Thus, SISALv1b
provides a sufficient coverage to allow the data to be used for model evaluation. The temporal
distribution of records is uneven, with only ca. 40 at 21 kyrs increasing to > 100 records at 6 kyrs and
> 110 for the last 1,000 yrs (Fig. 1b). A pronounced regional bias exists towards Europe during the
Holocene. Regional coverage is relatively even during the LGM, with the exception of Africa which is
under-represented throughout (< 4% of total). Nevertheless, there is sufficient coverage to facilitate
data-model comparisons for the MH and LGM for most regions of the world.





The global occurrence of speleothems through time approximates an exponential distribution (Fig. 2
a, b). Anomalously high numbers of speleothems are found in the last 12 kyrs, between 128–112 kyrs
BP and during interglacials MIS 1 and 5e (and the early glacial MIS 5d). There are fewer than expected
speleothems between 73–63 kyrs BP and during MIS 2. These deviations could arise from sampling
biases but may also reflect globally wetter or drier intervals. Differences between curves constructed
for tropical and temperate regions (Fig. 2 c, d) suggest these deviations are climatic in origin because
there is less variability in the tropical than the temperate curve. Thus, even at a global level, the
speleothem data provide a first-order assessment of climate changes on orbital time scales.
**3.2. How well do the speleothem records represent modern $\delta^{18}O$ in precipitation?**
The first-order spatial patterns shown by the SISAL speleothem records during the modern period
(1960–2017; n = 72) are in overall agreement with the OIPC dataset of interpolated $w\delta^{18}O_p$ ($R^2 = 0.78$),
with more negative values at higher latitudes and in more continental climates (Fig. 3a) as shown by
McDermott et al. (2011) for European stalagmites. Low latitude sites tend to show more positive $\delta^{18}O$
values than the OIPC data, whereas sites from the mid to high-latitudes tend to be more negative (Fig.
3b). A similar bias is observed in the comparison between SISAL and the simulated $w\delta^{18}O_p$ ($R^2 = 0.79$),
although in this case the slope is steeper (Fig. 3 c, d). Some discrepancies between the SISAL data and
the observations or simulations may be due to cave specific factors such as a preferred seasonality of
recharge, non-equilibrium fractionation processes during speleothem deposition or by complex soil-
atmosphere interactions affecting evapotranspiration and, thus, the isotopic signal of the effective
recharge. However, the overall level of agreement suggests that the SISAL data provide a good
representation of the impacts of modern hydroclimatic processes.
Comparison of the SISAL records with $\delta^{18}O_p$ weighted according to the potential recharge amount or
with $\delta^{18}O_{sw}$ weighted to the moisture amount does not significantly improve the data-model
comparison (Supplementary Fig. 1). The best relationship is obtained with $w\delta^{18}O_{sw}$ ($R^2 = 0.80$).
However, smoothing the simulated $w\delta^{18}O_p$ records on a sample-to-sample basis to account for multi-



year transit times in the karst environment produces a slightly better geographic agreement with the
SISAL records (Supplementary Fig. 2). Accounting for differences between the model grid cell and cave
elevations does not yield any overall improvement in the global correlations.
Simulated inter-annual variability is less than shown in the GNIP data (Fig. 4). Although there are
missing values for the GNIP station data, we have also removed these intervals from the simulations,
so incomplete sampling is unlikely to explain the difference between the observed and simulated
inter-annual variability. The inter-annual variability of the modern speleothem records is lower than
both the simulated and the GNIP data, reflecting the impact of within karst and cave processes that
effectively act as a low-pass filter on the signal recorded during speleothem growth. Smoothing the
simulated $\delta^{18}O_p$ signal produces a better match to the SISAL records: application of a smoothing
window of > 5 yrs to simulated w$\delta^{18}O_p$ produces a good match (95% confidence) with the inter-annual
variability shown by the speleothems (Fig. 4). The fact that the temporal smoothing of the simulations
produces a better match both in terms of geographic patterns and inter-annual variability results from
the tendency of speleothem records to predominantly contain low-frequency information (Baker et
al., 2013) and indicates that data-model comparisons using speleothem records should focus on quasi-
decadal or longer timescales.
**3.3. Anomaly-mode time-slice comparisons**
The selection of a modern or pre-industrial base period is a first step in reconstructing speleothem
$\delta^{18}O$ anomalies for comparisons with simulated changes in specific model experiments. There are 62
speleothem records that cover the pre-industrial interval 1850±15 CE, commonly used as a reference
in model experiments. However, using this short interval as the base period for comparisons with MH
or LGM simulations would result in the reconstruction of anomalies for only 18 records for the MH
and only 5 records for the LGM - which are the number of speleothem records with isotopic samples
in both the base period and either the MH or LGM (Table 2). There is no significant difference in the
mean $\delta^{18}O$ values for this pre-industrial period and the modern $\delta^{18}O$ values ($R^2$ = 0.96; Supplementary



Fig. 3). Using an extended modern baseline (1850–1990 CE) increases the data uncertainties by only
±0.5‰ but raises the number of MH records for which MH-modern anomalies can be calculated to 34
entities from 29 sites around the world. There is also an improvement in the number of LGM sites for
which it is possible to calculate anomalies, from 5 to 11 entities at 10 sites. Although longer base
periods have been used for data-model comparisons, for example the last 1,000 years (e.g., Werner
et al., 2016), this would increase the uncertainties in the observations without substantially increasing
the number of records for which it would be possible to calculate anomalies, particularly for the LGM
(Table 2). We, therefore, recommend the use of the interval 1850–1990 CE as the baseline for
calculation of $\delta^{18}O$ anomalies from the speleothem records.
A relatively good agreement exists between the sign of the simulated and observed $\delta^{18}O$ changes at
the MH and LGM: 77% of the MH entities and 64% of the LGM entities show changes in the same
direction after allowing for an uncertainty of ±0.5‰ (Fig. 5 a, b). However, the magnitude of the
changes is larger in the SISAL records than the simulations. The MH-modern speleothem anomalies
range from -3.63 to 1.28‰ (mean±std: -0.58±1.01‰), but the simulated anomalies only range from -
1.03 to 0.30‰ (mean±std: -0.13±0.31‰). Observed anomalies are 5–20 times larger than simulated
anomalies in the Asian monsoon region, and in individual sites in North and South America and
Uzbekistan (Fig. 5 a). The data-model mismatch is smallest in Europe, with a mean offset of
0.24±0.40‰ (n = 9 entities from 7 sites). Multivariate analyses (Supplementary Information) also show
that there is no significant relationship between observed and simulated $\delta^{18}O$ patterns in the MH. A
two-tailed Student t-test shows that most of the simulated anomalies are not significantly different
from present (at 95% confidence). This may reflect the fact that the *midHolocene* simulation was only
run for 10 years but is also consistent with previous studies which show that climate models
substantially underestimate the magnitude of MH changes (Harrison et al., 2014), particularly in
monsoon regions (e.g., Perez-Sanz et al., 2014).



The simulated changes in δ¹⁸O at the LGM are much larger than those simulated for the MH and are
significant (at 95% confidence) over much of the globe. There is no regionally coherent pattern in the
observed LGM anomalies because of the limited number of speleothems that grew continuously from
the LGM to present. However, the sign of the observed changes is coherent with the simulated change
in δ¹⁸O for 7 of the 11 records (Fig. 5 b). The magnitude of the LGM anomalies differs by less than 1‰
between model and data in half of the locations. A strong offset is found in the two records from
Sofular Cave, which are ca. 6‰ more negative than the simulated δ¹⁸O. This offset may be related to
the glacial changes in the Black Sea region, which are not well represented in the *lgm* simulation. Thus,
although overall the comparison with the speleothem records suggests that the simulated changes in
hydroclimate are reasonable, the simulated changes in the Middle East differ from observations.
However, multivariate analyses (Supplementary Information) reveal no significant relationship
between observed and simulated LGM δ¹⁸O patterns.
An alternative approach to examine the realism of simulated changes is to compare the LGM and MH
simulations directly, which improves the number of records for which anomalies can be calculated
(Fig. 5 c; n = 20). However, the pattern of change is similar to the LGM-modern anomalies. The
simulated and observed direction of change is coherent at 80% of the locations with an offset smaller
than 1‰ occurring in 7 sites and again the largest discrepancy is Sofular Cave. Thus, in this particular
example, a direct comparison of the LGM-MH anomalies does not provide additional insight to the
comparison of LGM-modern anomalies. Nevertheless, such an approach might be useful for other
time periods (e.g., comparison of early versus mid-Holocene) when there are likely to be many more
speleothem records available.
Age uncertainties inherent to the speleothem samples selected to represent the LGM could partially
explain the LGM data-model mismatches. A global assessment of the impact of time-window width
on the MH and LGM anomalies shows that reducing the window width from ±500 to ±200 years in the
MH has little impact on the average values (Supplementary Fig. 4) but reduces the inter-sample



variability and produces a better match to the simulated anomalies. A similar analysis for the LGM
(Supplementary Fig. 5) suggests that a window-width of ±500 years (rather than ±1,000 years) would
be the most appropriate choice for comparisons of this interval. The number of SISAL sites available
for such comparisons is not affected. However, analyses of the relative error of the isotope anomalies
calculated at individual sites for different LGM window widths (Fig. 6) show a clear increase in all
relative error components as window size decreases for BT-2 but no clear changes in the relative error
terms for SSC01 (the samples from Botuverá and Gunung-buda cave, respectively, with new COPRA-
produced age-depth models). These results suggest that, with an LGM window width of ±1,000 years,
the relative contribution of age uncertainty to the anomaly uncertainty is small (Fig. 6). Thus, although
it is clear that it would be useful to propagate age uncertainties for individual sites, changing the
conventional definitions of the MH and LGM time slices in deriving speleothem anomalies does not
seem warranted at this stage.

### 3.4. Analysis of spatial gradients

The number of sites available in SISALv1b means that quantitative data-model comparisons using the
traditional anomaly approach are limited in scope. Approaches based on comparing trends in absolute
$\delta^{18}O$ values could provide a way of increasing the number of observations and an alternative way to
evaluate the simulations. Comparison of trends places less weight on anomalous sites and allows
large-scale systematic similarities and dissimilarities between model and observations to be revealed.
We illustrate this approach using spatial gradients in the MH and LGM, although such an approach
could also be used for temporal trends.
The first-order trends in observed $\delta^{18}O$ changes during the MH are broadly captured by the model
(Fig. 7). The largest mismatches between the observations and simulations, in the high latitudes of
North America, in mid-latitude Europe and in the monsoon region of Asia, are in regions where the
model does not reflect the reconstructed MAP. This confirms the suggestion, based on comparison of
the MH mapped patterns (section 3.3), that ECHAM5-wiso underestimates changes in precipitation



between the MH and the present day. The observed latitudinal $\delta^{18}O$ gradients in the LGM are
reasonably well captured by the simulations (Fig. 8), reflecting the strong latitudinal control on $\delta^{18}O$
variability (Dansgaard, 1964). As is the case in the MH, the largest discrepancies occur in regions where
the model overestimates MAP. However, this mismatch may partly reflect the fact that the pollen-
based reconstructions do not take account of the low atmospheric $CO_2$ concentration during the glacial
and, may consequently underestimate the actual precipitation amount (Prentice et al., 2017).
Nevertheless, these examples show the potential to use trends in absolute values for model evaluation
and diagnosis.
**4.   Protocol for data-model comparison using speleothem data**
Our analyses illustrate a number of possible approaches for utilising use speleothem isotopic data
towards model evaluation. The discontinuous nature of most speleothem records means that the
number of sites available for conventional anomaly-mode comparisons is potentially limited. To some
extent this is mitigated by the fact that differences between the modern and pre-industrial isotope
values are small, permitting the calculation of anomalies using a longer baseline interval (1850–1990
CE). The use of smaller intervals of time in calculating MH or LGM anomalies (Supplementary Fig. 4
and 5) does not have a significant impact either on the mean values or the number of records provided
the interval is > ±300 yrs for the MH and > ±500 yrs for the LGM. Although the use of shorter intervals
is possible, we recommend using the conventional definitions of each time slice to facilitate
comparison with other benchmark datasets. Although patterns in the isotopic anomalies can provide
a qualitative assessment of model performance, site-specific factors could lead to large differences
from the simulations at individual locations. Improved spatial coverage would allow such sites to be
identified and screened out before making quantitative comparisons of observed and simulated
anomalies. More records are available for the MH or LGM alone than for both that period (i.e. MH or
LGM) and the modern baseline period, encouraging examination of spatial gradients in absolute $\delta^{18}O$.
Even when an offset between the observed and simulated $\delta^{18}O$ exists, comparing the trends along



such gradients is possible. Thus, both absolute values and anomalies of the isotope data for data-
model comparison are useful.
Screening of published speleothem isotopic data is essential to produce meaningful data-model
comparisons. The SISAL database facilitates screening for mineralogy, which has a substantial effect
on isotopic values because of differences in water-carbonate fractionation factors for aragonite or
calcite. We recommend the use of the empirical speleothem-based fractionation factor of Tremaine
et al. (2011) for isotope values on calcite stalagmites, or on aragonite specimens that have been
corrected to their calcite equivalent in the original publications, and the equilibrium fractionation
equation of Grossman and Ku (1986) reformulated in Lachniet (2015) for aragonite samples to ensure
consistency across records.
Based on the limited number of records available at the LGM, speleothem age uncertainties have only
a limited impact on mean isotopic values, propagation of such uncertainties as well as any model
uncertainties would substantially improve the robustness of data-model comparisons.
Based on our analyses, we therefore recommend that model evaluation using speleothem records
should:
1. Filter speleothem records with respect to their mineralogy and use the appropriate equilibrium

fractionation factor: Tremaine et al. (2011) for converting isotopic data from either calcite or

aragonite-corrected-to-calcite samples to their drip water equivalent; and Grossman and Ku

(1986) as reformulated by Lachniet (2015) for converting isotopic data from aragonite samples;

2. use the interval between 1850 and 1990 as the reference period for speleothem isotope records;
3. use speleothem isotopic data averaged for the intervals 6,000 ±500 yrs (21,000 ±1,000 yrs) for

comparability with other MH (LGM) palaeoclimate benchmark datasets;

4. use speleothem isotopic data averaged for the interval 6,000 ±200 yrs or 21,000 ±500 yrs for best

approximation of *midHolocene* and *lgm* experiments;



5. use absolute values only to assess data-model first order spatial patterns;
6. focus on multi-decadal to millennial timescales if using transient simulations for data-model

comparisons.

**5.   Conclusions**
Speleothem records show the same first-order spatial patterns as available in the Global Network of
Isotopes in Precipitation (GNIP) data, and, therefore, are a good reflection of the $\delta^{18}$O patterns in
modern precipitation. This observation then suggests that stalagmites are a rich source of information
for model evaluation. However, the inter-annual variability in the modern speleothem records is
considerably reduced compared to the simulations, which in turn show less inter-annual variability
than the GNIP observations. The low variability shown by the SISAL records – most likely from the low-
pass filter effectively applied to the speleothem record by the karst system – precludes the use of this
database for global studies focused on time scales shorter than quasi-decadal on a global basis.
Using the traditional anomaly approach to data-model comparisons, consistency between the sign of
observed and simulated changes in both the MH and the LGM exists. However, the amplitude of
modelled $\delta^{18}$O changes is lower than the amplitude observed in the speleothem records. Thus, these
kinds of comparisons should only focus on the large-scale spatial patterns that are significant, robust
and climatologically interpretable. Based on the available SISAL data, the use of smaller time windows
than the conventional definitions for each time slice does not have a strong impact on the mean values
and could be used to reduce the uncertainties associated with the palaeodata. However, this would
preclude comparisons with existing benchmark datasets that use the conventional windows for the
MH and LGM time slices.
Only a limited number of speleothem records are continuous over long periods of time and the need
to convert these to anomalies with respect to modern is a drawback. The limited number of records
covering the LGM make the comparisons for this period particularly challenging. Nevertheless,



continued expansion of SISAL database will increase its usefulness for model evaluation in future.
Furthermore, we have shown that alternative approaches using absolute values could help examine
spatial trends and diagnose systematic offsets.
Difficulties in constraining structural error on the model side and local controls on the observations
complicate the derivation of comprehensive estimates of the true uncertainties of both simulations
and observations. Site-specific controls can affect the $\delta^{18}O$ record captured in speleothems, but we
have not screened for regionally anomalous records that could be influencing the results in our
analyses. Our initial analyses suggest age uncertainty contributes little to the estimates for the LGM
speleothem isotopic values. However, it is still important to propagate dating uncertainties for data-
model comparison. Despite these challenges, SISAL appears to be an extremely useful tool for
describing past patterns of variability, highlighting its potential for evaluating CMIP6-PMIP4
experiments.
**6. Data availability**
The version of the SISAL database used in this study is available in the University of Reading
Research Data Archive (http://dx.doi.org/10.17864/1947.189). This dataset is cited in this
manuscript as Atsawawaranunt et al., 2019.
**7. Team list**
SISAL working group members who coordinated data gathering for the SISAL database (listed in
alphabetical order): Syed Masood Ahmad (Department of Geography, Faculty of Natural Sciences,
Jamia Millia Islamia, New Delhi 110025, India), Yassine Ait Brahim (Institute of Global Environmental
Change, Xi'an Jiaotong University, Xi'an, Shaanxi, China), Sahar Amirnezhad Mozhdehi (School of Earth
Sciences, University College Dublin, Belfield. Dublin 4, Ireland), Monica Arienzo (Division of Hydrologic
Sciences, Desert Research Institute, 2215 Raggio Parkway, 89512 Reno, NV, USA), Kamolphat
Atsawawaranunt (School of Archaeology, Geography & Environmental Sciences, Reading University,

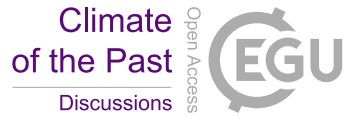

Whiteknights, Reading, RG6 6AH, UK), Andy Baker (School of Biological, Earth and Environmental
Sciences, University of New South Wales, Kensington 2052, Australia), Kerstin Braun (Institute of
Human Origins, Arizona State University, PO Box 874101, 85287 Tempe, Arizona, USA), Sebastian
Breitenbach (Sediment & Isotope Geology, Institute of Geology, Mineralogy & Geophysics, Ruhr-
Universität Bochum, Universitätsstr. 150, IA E5-179, 44801 Bochum, Germany), Yuval Burstyn
(Geological Survey of Israel, 32 Yesha'yahu Leibowitz, 9371234, Jerusalem, Israel; Institute of Earth
Sciences, Hebrew University of Jerusalem, Edmond J. Safra campus, Givat Ram, 91904 Jerusalem,
Israel), Sakonvan Chawchai (MESA Research unit, Department of Geology, Faculty of Sciences,
Chulalongkorn University, 254 Phayathai Rd, Pathum Wan, 10330 Bangkok, Thailand), Andrea
Columbu (Department of Biological, Geological and Environmental Sciences, Via Zamboni 67, 40126,
Bologna, Italy), Michael Deininger (Institute of Geosciences, Johannes Gutenberg University Mainz,
Johann-Joachim-Becher-Weg 21, 55128 Mainz, Germany), Attila Demény (Institute for Geological and
Geochemical Research, Research Centre for Astronomy and Earth Sciences, Hungarian Academy of
Sciences, Budaörsi út 45, H-1112 Budapest, Hungary), Bronwyn Dixon (School of Archaeology,
Geography & Environmental Sciences, Reading University, Whiteknights, Reading, RG6 6AH, UK and
School of Geography, University of Melbourne, Melbourne, 3010, Australia), István Gábor Hatvani
(Institute for Geological and Geochemical Research, Research Centre for Astronomy and Earth
Sciences, Hungarian), Jun Hu (Department of Earth Sciences, University of Southern California, 3651
Trousdale Parkway, 90089 Los Angeles, California, USA), Nikita Kaushal (Department of Earth Sciences,
University of Oxford, South Parks Road, Oxford, OX1 3AN, UK), Zoltán Kern (Institute for Geological
and Geochemical Research, Research Centre for Astronomy and Earth Sciences, Hungarian Academy
of Sciences, Budaörsi út 45, H-1112 Budapest, Hungary), Inga Labuhn (Institute of Geography,
University of Bremen, Celsiusstr. 2, 28359 Bremen, Germany), Matthew S. Lachniet (Dept. of
Geoscience, University of Nevada Las Vegas, Box 4022, 89154 Las Vegas, NV, USA), Franziska A.
Lechleitner (Department of Earth Sciences, University of Oxford, South Parks Road, OX1 3AN Oxford,
UK), Andrew Lorrey (National Institute of Water & Atmospheric Research, Climate Atmosphere and



Hazards Centre, 41 Market Place, Viaduct Precinct, Auckland, New Zealand), Monika Markowska
(University of Tübingen, Hölderlinstr. 12, 72074 Tübingen, Germany), Carole Nehme (IDEES UMR 6266
CNRS, Geography department, University of Rouen Normandie, Mont Saint Aignan, France), Valdir F.
Novello (Instituto de Geociências, Universidade de São Paulo, São Paulo, Brazil), Jessica Oster
(Department of Earth and Environmental Sciences, Vanderbilt University, Nashville, TN, 37206, USA),
Carlos Pérez-Mejías (Department of Geoenvironmental Processes and Global Change, Pyrenean
Institute of Ecology (IPE-CSIC), Avda. Montañana 1005, 50059 Zaragoza, Spain), Robyn Pickering
(Department of Geological Sciences, University of Cape Town, University Avenue, Upper Campus,
Room 208, 7701 Rondebosch, Cape Town, South Africa), Natasha Sekhon (Department of Geological
Sciences, Jackson School of Geosciences, University of Texas, Austin, TX, 78712, USA), Xianfeng Wang
(Earth Observatory of Singapore, Nanyang Technological University, Singapore 636798), Sophie
Warken (Institute of Environmental Physics, Ruprecht-Karls-Universität Heidelberg, Im Neuenheimer
Feld 229, 69120 Heidelberg, Germany.
SISAL working members who submitted data to the SISAL database (listed in alphabetical order): Tim
Atkinson (Departments of Earth Sciences & Geography, University College London, WC1E 6BT, United
Kingdom), Avner Ayalon (Geological Survey of Israel, 32 Yesha'yahu Leibowitz, Jerusalem, 9371234),
James Baldini (Department of Earth Sciences, Durham University, DH1 3LE, United Kingdom), Miryam
Bar-Matthews (Geological Survey of Israel, 32 Yesha'yahu Leibowitz, Jerusalem, 9371234), Juan Pablo
Bernal (Centro de Geociencias, Universidad Nacional Autónoma de México, Campus UNAM Juriquilla,
Querétaro 76230, Querétaro, Mexico), Ronny Boch (Graz University of Technology.  Institute of
Applied Geosciences.  Rechbauerstrasse 12, 8010 Graz, Austria), Andrea Borsato (School of
Environmental and Life Science, University of Newcastle, 2308 NSW, Australia), Meighan Boyd
(Department of Earth Sciences, Royal Holloway University of London, Egham, Surrey TW20 0EX, UK),
Chris Brierley (Department of Geography, University College London, WC1E 6BT, United Kingdom),
Yanjun Cai (State Key Lab of Loess and Quaternary Geology, Institute of Earth Environment, Chinese
Academy of Sciences, Xi'an 710061, China), Stacy Carolin (Institute of Geology, University of Innsbruck,





Innrain 52, 6020 Innsbruck, Austria), Hai Cheng (Institute of Global Environmental Change, Xi'an
Jiaotong University, China), Silviu Constantin (Emil Racovita Institute of Speleology, str. Frumoasa 31,
Bucharest, Romania), Isabelle Couchoud (EDYTEM, UMR 5204 CNRS, Université Savoie Mont Blanc,
Université Grenoble Alpes, 73370 Le Bourget du Lac, France), Francisco Cruz (Instituto de Geociências,
Universidade de São Paulo, São Paulo, Brazil), Rhawn Denniston (Department of Geology, Cornell
College, Mount Vernon, IA, 52314, USA), Virgil Drăguşin (Emil Racovita Institute of Speleology, str.
Frumoasa 31, Bucharest, Romania), Wuhui Duan (Key Laboratory of Cenozoic Geology and
Environment, Institute of Geology and Geophysics, Chinese Academy of Sciences, Beijing, China,
100029, Vasile Ersek (Department of Geography and Environmental Sciences, Northumbria University,
Newcastle upon Tyne, UK), Martin Finné (Department of Archaeology and Ancient History,  Uppsala
University, Sweden), Dominik Fleitmann (Department of Archaeology. School of Archaeology,
Geography and Environmental Science. Whiteknights. University of Reading RG6 6AB, UK), Jens
Fohlmeister (Institute for Earth and Environmental Sciences, University of Potsdam, Karl-Liebknecht-
Str. 24-25, 14476 Potsdam, Germany), Amy Frappier (Department of Geosciences, Skidmore College,
Saratoga Springs, NY 12866 US), Dominique Genty (Laboratoire des Science du Climat et de
l'Environment, CNRS, L'Orme des Merisiers, 91191 Gif-sure-Yvette Cedex, France), Steffen
Holzkämper (Department of Physical Geography. Stockholm University. 106 91 Stockholm, Sweden),
Philip Hopley (Department of Earth and Planetary Sciences, Birkbeck, University of London, Malet St,
London, WC1E 7HX), Vanessa Johnston (Karst Research Institute, Research Centre of the Slovenian
Academy of Sciences and Arts, Titov trg 2, SI-6230, Postojna, Slovenia), Gayatri Kathayat (Institute of
Global Environmental Change, Xi'an Jiaotong University, China), Duncan Keenan-Jones (School of
Historical and Philosophical Inquiry, University of Queensland, St Lucia QLD 4072, Australia), Gabriella
Koltai (Institute of Geology, University of Innsbruck, Innrain 52, 6020 Innsbruck, Austria), Ting-Yong Li
(Chongqing Key Laboratory of Karst Environment, School of Geographical Sciences, Southwest
University, Chongqing 400715, China, and Field Scientific Observation & Research Base of Karst Eco-
environments at Nanchuan in Chongqing, Ministry of Nature Resources of China, Chongqing 408435,



China), Marc Luetscher (Swiss Institute for Speleology and Karst Studies (SISKA), Rue de la Serre 68,
CH-2301 La Chaux-de-Fonds, Switzerland), Dave Mattey (Department of Earth Sciences, Royal
Holloway University of London, Egham, Surrey, TW20 0EX, UK), Ana Moreno (Dpto. de Procesos
Geoambientales y Cambio Global. Instituto Pirenaico de Ecología-CSIC. Zaragoza, Spain), Gina Moseley
(Institute of Geology, University of Innsbruck, Innrain 52, 6020 Innsbruck, Austria), David Psomiadis
(Imprint Analytics GmbH, Werner von Siemens Str. 1, A7343 Neutal, Austria), Jiaoyang Ruan
(Guangdong Provincial Key Lab of Geodynamics and Geohazards, School of Earth Sciences and
Engineering, Sun Yat-sen University, Guangzhou 510275, China), Denis Scholz (Institute for
Geosciences, University of Mainz, Johann-Joachim-Becher-Weg 21, 55128 Mainz, Germany), Lijuan
Sha (Institute of Global Environmental Change, Xi'an Jiaotong University, Xi'an, Shaanxi, China), James
Baucher 5, Sofia 1164, Bulgaria), Andrew Christopher Smith (NERC Isotope Geoscience Facility, British
Geological Survey, Nottingham, UK), Nicolás Strikis (Departamento de Geoquímica, Universidade
Federal Fluminense, Niterói, Brazil), Pauline Treble (ANSTO, Lucas Heights NSW, Australia), Ezgi Ünal-
İmer (Department of Geological Engineering, Middle East Technical University, Ankara, Turkey ), Anton
Vaks (Geological Survey of Israel, 32 Yesha'yahu Leibowitz, Jerusalem, 9371234), Stef Vansteenberge
(Analytical, Environmental & Geo-Chemistry, Department of Chemistry, Vrije Universiteit Brussel,
Belgium), Ny Riavo G. Voarintsoa (Institute of Earth Sciences, The Hebrew University in Jerusalem,
Israel), Corinne Wong (Environmental Science Institute, The University of Texas at Austin, 2275
Speedway, Austin TX 78712, USA), Barbara Wortham (Department of Earth and Planetary Science,
University of California, Davis, USA), Jennifer Wurtzel (Research School of Earth Sciences, Australian
National University, Canberra, ACT, Australia / ARC Centre of Excellence for Climate System Science,
Australian National University, Canberra, ACT, Australia), Haiwei Zhang (Institute of Global
Environmental Change, Xi'an Jiaotong University, China).
**8.  Author contribution**





LCB is the coordinator of the SISAL working group. LCB and SPH designed the study. LCB and SPH wrote
the first draft of the manuscript with contributions from MW, NS, KR, CVP. LCB did the analyses and
created Figs. 1, 2-5, 7, 8 and Supplementary Figs. 1-5. MW provided the ECHAM5-wiso model
simulations and helped on its analyses. NS did the analyses on speleothem growth over time and
created Fig. 2. KR did the analysis on the LGM uncertainties and created Fig. 6. CVP did the multivariate
linear analyses (Supplementary material). All authors contributed to the last version of this
manuscript. The authors listed in the "SISAL working group" team contributed to this study
coordinating data gathering, database construction or with speleothem data submitted to the SISAL
database. SB created the COPRA age-depth models used in this study. TA and DG contributed original
unpublished data to the SISAL database. JB, AB, ZK, MA, MSL, VJ, BW and SB helped edit the
manuscript.
**9.   Competing interests**
The authors declare that they have no conflict of interest.
**10. Acknowledgements**
SISAL (Speleothem Isotopes Synthesis and Analysis) is a working group of the Past Global Changes
(PAGES) programme. We thank PAGES for their support for this activity. Additional financial support
for SISAL activities has been provided by the European Geosciences Union (EGU TE Winter call, grant
number W2017/413), Irish Centre for Research in Applied Geosciences (iCRAG), European Association
of Geochemistry (Early Career Ambassadors program 2017), Quaternary Research Association UK,
Navarino Environmental Observatory, Stockholm University, Savillex, John Cantle, University of
Reading (UK), University College Dublin (Ireland; Seed Funding award, grant number SF1428), and
University Ibn Zohr (Morocco). L.C.B. and S.P.H. acknowledge support from the ERC-funded project
GC2.0 (Global Change 2.0: Unlocking the past for a clearer future, grant number 694481). S.P.H. also
acknowledges support from the JPI-Belmont project "PAleao-Constraints on Monsoon Evolution and



Dynamics (PACMEDY)" through the UK Natural Environmental Research Council (NERC). L.C.B. also
acknowledges support from the Geological Survey Ireland (Short Call 2017; grant number 2017-SC-
056) and the Royal Irish Academy's Charlemont Scholar award 2018. C.V.P. acknowledges funding
from the Portuguese Science Foundation (FCT) through the CIMA research center project
(UID/MAR/00350/2013). K.R. was supported by Deutsche Forschungsgemeinschaft (DFG) grant no.
RE3994/2-1. We thank the World Karst Aquifer Mapping project (WOKAM) team for providing us with
the karst map presented in Fig. 1a. The authors would like to thank the following data contributors:
Dominique Blamart, Jean Riotte, Russell Drysdale, Petra Bajo and Frank McDermott.

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





**Figure Captions**

**Figure 1:** Spatio-temporal distribution of SISALv1b database. **(a)** Spatial distribution of speleothem
records. Filled circles are sites used in this study (SISALv1 in purple; SISALv1b in light blue). Crosses are
SISAL sites that do not pass the screening described in section 2.3 and/or do not cover the time periods
used here (modern, MH and LGM). The background carbonate lithology is that of the World Karst
Aquifer Mapping (WOKAM) project (Chen et al., 2017). **(b)** Temporal distribution of speleothem
records according to regions. The non-overlapping bins span 1,000 years and start on 1950 CE. Regions
have been defined as: Oceania (-60° < Lat < 0°; 90° < Lon < 180°); Asia (0° < Lat < 60°; 60° < Lon < 130°);
Middle East (7.6° < Lat < 50°; 26° < Lon < 59°); Africa (-45° < Lat < 36.1°; -30° < Lon < 60°; with records
in the Middle East region removed); Europe (36.7° < Lat < 75°; -30° < Lon < 30°; plus Gibraltar and
Siberian sites); South America (-60° < Lat < 8°; -150° < Lon < -30°); North and Central America (8.1° <
Lat < 60°; -150° < Lon < -50°).

**Figure 2:** Distribution of the number of unique caves with speleothem growth through time. **(a)**
Number of unique caves with growth over the last 500 yrs BP in 1000-year bins (solid line),
bootstrapped estimate of uncertainty (shading between 5 and 95% percentiles) and fitted exponential
distribution (darker solid line). **(b, c)** same as a) but with the fitted exponential distribution subtracted
to highlight anomalies from the expected number of caves over the last 300 kyrs BP. Horizontal bars
in b) and c) indicate periods with significantly greater (dark grey) or fewer (light grey) number of caves
with speleothem growth than expected. Green indicates the full global dataset, blue and red indicate
temperate and tropical subdivisions respectively. Horizontal bars in a) denote previous interglacials.

**Figure 3:** Comparison of SISAL data with observational and simulated w$\delta^{18}$O$_p$ for the modern
period. **(a)** Comparison between SISAL $\delta^{18}$O averages [‰; V-SMOW] for the period 1960–2017 with
OIPC data [‰; V-SMOW]. **(b)** Scatterplot of SISAL modern $\delta^{18}$O averages as in (a) versus w$\delta^{18}$O$_p$
extracted from OIPC (i.e., background map in (a)) at the location of each cave site. **(c)** Same than (a)
with simulated w$\delta^{18}$O$_p$ data for the period 1958–2013 in the background. **(d)** Scatterplot of SISAL





modern $\delta^{18}$O as in (c) versus the simulated w$\delta^{18}$O$_p$ for the period 1958–2013. Dashed lines in (b) and
(d) represents the 1:1 line.  All SISAL isotope data have been converted to their drip-water equivalent
using the calcite-water $\delta^{18}$O fractionation equation from Tremaine et al. (2011). Mean annual air
surface temperature from CRU-TS4.01 (Harris et al., 2014) and mean annual simulated ECHAM5-wiso
air surface temperature were used as surrogates for cave temperatures in the OIPC and ECHAM5-wiso
comparison, respectively. See section 2.3 for details on data extraction and conversion.
**Figure 4:** Modern global inter-annual $\delta^{18}$O variability. Box plots show the variability of the standard
deviation of global annual $\delta^{18}$O using: **(left)** GNIP stations with at least 10 months of data per year and
at least 5 years of data (n = 450) and ECHAM5-wiso data extracted at the location of each GNIP station
for the years when this data is available; **(right)** SISAL records with at least 5 isotope samples for the
period 1958–2013 and simulated w$\delta^{18}$O$_p$ extracted at each cave location for the same years for which
speleothem data is available. Boxplots in shades of red at the rightmost of the panel are constructed
after smoothing the simulated w$\delta^{18}$O$_p$ data for 1 to 16 years . On each box, the central red mark
indicates the median (q$_2$; 50$^{th}$ percentile) and the bottom and top edges of the box indicate the 25$^{th}$
(q$_1$) and 75$^{th}$ (q$_3$) percentiles, respectively. Outliers (red crosses) are locations with standard deviations
greater than q$_3$ + 1.5 × (q$_3$ - q$_1$) or less than q$_1$ - 1.5 × (q$_3$ - q$_1$). This corresponds to approximately ±2.7σ
or 99.3% coverage if the data are normally distributed. If the notches in the box plots do not overlap,
you can conclude, with 95% confidence, that the true medians do differ. The grey horizontal band
corresponds to the notch in SISAL for easy comparison. SISAL were converted to their drip-water $\delta^{18}$O
equivalent as described in section 2.3.
**Figure 5:** ECHAM5-wiso weighted $\delta^{18}$O$_p$ anomalies ([‰; V-SMOW]; background map) and SISAL
isotope anomalies ([‰; V-PDB]; filled circles) for three time-slices: **(a)** MH-PI (SISAL records n = 34),
**(b)** LGM-PI (SISAL records n = 11) and **(c)** LGM-MH (SISAL records n = 20). For easy visualisation, when
there are two speleothem records from the same cave site, one has been shifted 2$^o$ towards the North
and the East (shown here as triangles). Note the different colour bar axis in the colour bar of (a)



compared to (b) and (c). Two-tailed student t-test has been applied to calculate the significance of the
ECHAM5-wiso anomalies in (a) and (b) at a 95% confidence. No significance has been calculated for
(c), which compares two different simulations with their corresponding control periods. SISAL
anomalies calculated with respect to 1850–1990. SISAL data has been converted to its drip water
equivalent prior to calculating the anomalies.
**Figure 6**: LGM period definitions and their impact on SISAL $\delta^{18}O$ mean estimate uncertainty. The
impact of the window definition and age uncertainty is explored for two entities **(a)** entity BT-2 from
Botuverá cave (Cruz et al., 2005)and **(b)** entity SSC01 from Gunung-buda cave (Partin et al., 2007). The
relative error is defined as 2 standard deviations for the original age model and the COPRA median;
and the upper minus lower 95% quantiles for the COPRA median uncertainty as well as the COPRA
ensemble spread of standard deviations. Black solid lines give the relative error of the mean isotopic
estimate for the LGM for the original age model, the grey solid line for the estimate based on the
COPRA median age model. The pink dotted line shows the uncertainty of the COPRA median estimate,
and the green dashed line the average relative error estimate across the 1,000-member COPRA
ensemble. For both speleothems, relatively stable error estimates are found for window sizes larger
than 750 years, whereas the relative error increases towards smaller window sizes.
**Figure 7:** Mid-Holocene (MH) transects for three regions: **(a)** NW to SE across North America; **(b)**
N-S across southern Europe and northern Africa, and **(c)** NW to SE across SE Asia. Maps at the top of
each panel show the simulated $\delta^{18}O_p$ (left), Mean Annual Temperature (MAT; centre) and Mean
Annual Precipitation (MAP; right) from ECHAM5-wiso. The same scale is used for the $\delta^{18}O$, MAT and
MAP maps. All transects show absolute $\delta^{18}O$ values. In the $\delta^{18}O$ maps, filled circles are SISAL $\delta^{18}O$
averages for entities that cover both the MH and the modern reference period. Filled squares are
SISAL entities that do not have a corresponding modern. Bottom plots of each panel show the
simulated data extracted for each transect: black circles and whiskers are mean ±1 standard deviation
of the data extracted along longitudinal sections in between the two great circle lines shown in solid



grey lines in the top maps. The red line is the median of the extracted data. All data were extracted at
steps of 1.12° to coincide with the average model grid-size.  Bottom plots in each panel also show
SISAL δ$^{18}$O (circles for low-elevation sites, < 1,000 masl; triangles for high-elevation sites, > 1,000 masl),
pollen-based quantitative reconstructions of MAT (red squares; Bartlein et al., 2011) and MAP (blue
squares; Bartlein et al., 2011). Pollen-based reconstructions have been converted to absolute values
by adding the CRU-TS4.01 climatology (Harris et al., 2014).
**Figure 8:** Last Glacial Maximum (LGM) transects for three regions: **(a)** NW to SE across North
America; **(b)** N-S from central Europe to southern Africa, and **(c)** NW-SE from China to northern
Australia. Details as in caption of Fig. 7.
**Table Captions**
**Table 1:** List of speleothem records that have been added to SISALv1 (Atsawawaranunt et al.,
2018a;Atsawawaranunt et al., 2018b) to produce SISALv1b (Atsawawaranunt et al., 2019) sorted
alphabetically by site name. Elevation is in metres above sea level (masl), latitude in degrees North
and longitude in degrees East.
**Table 2:** Number of SISALv1b speleothem records available for key time periods. Mid-Holocene
(MH): 6±0.5 kyrs BP; Last Glacial Maximum (LGM): 21±1 kyrs BP. "kyrs BP" refers to thousand years
before present, where present is 1950 CE.



**Figure 1**

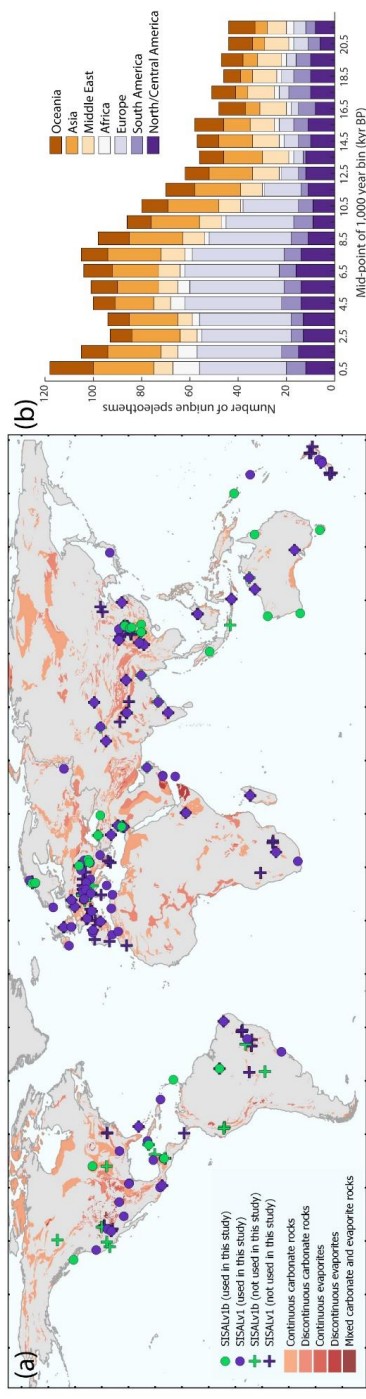



**Figure 2**

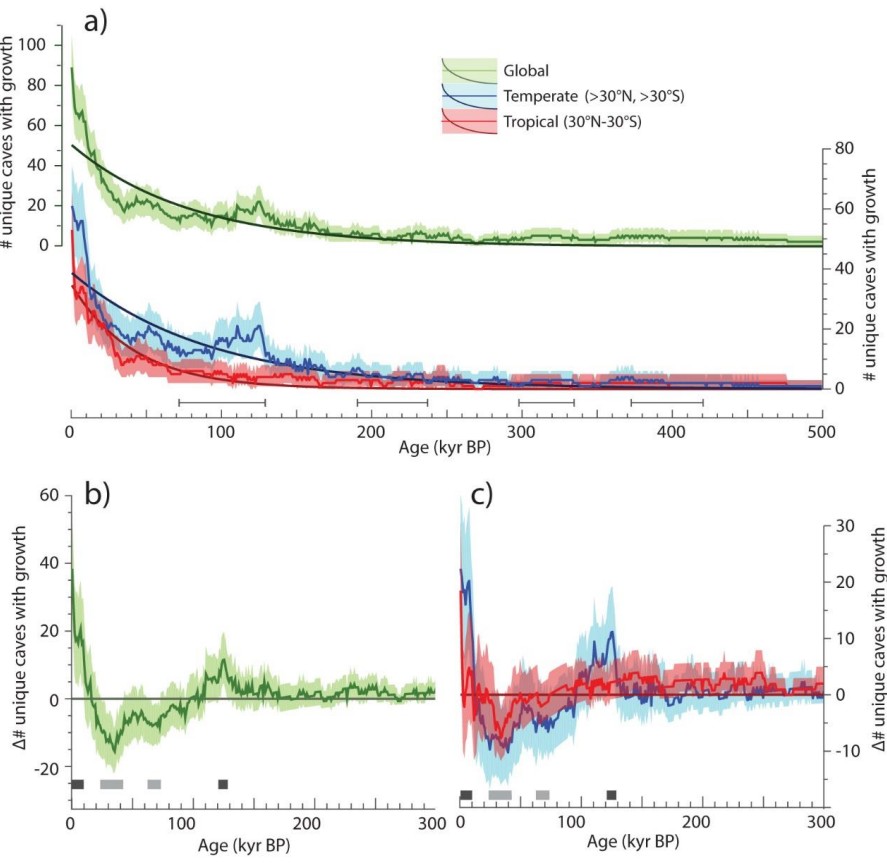





**Figure 3**





**Figure 4**

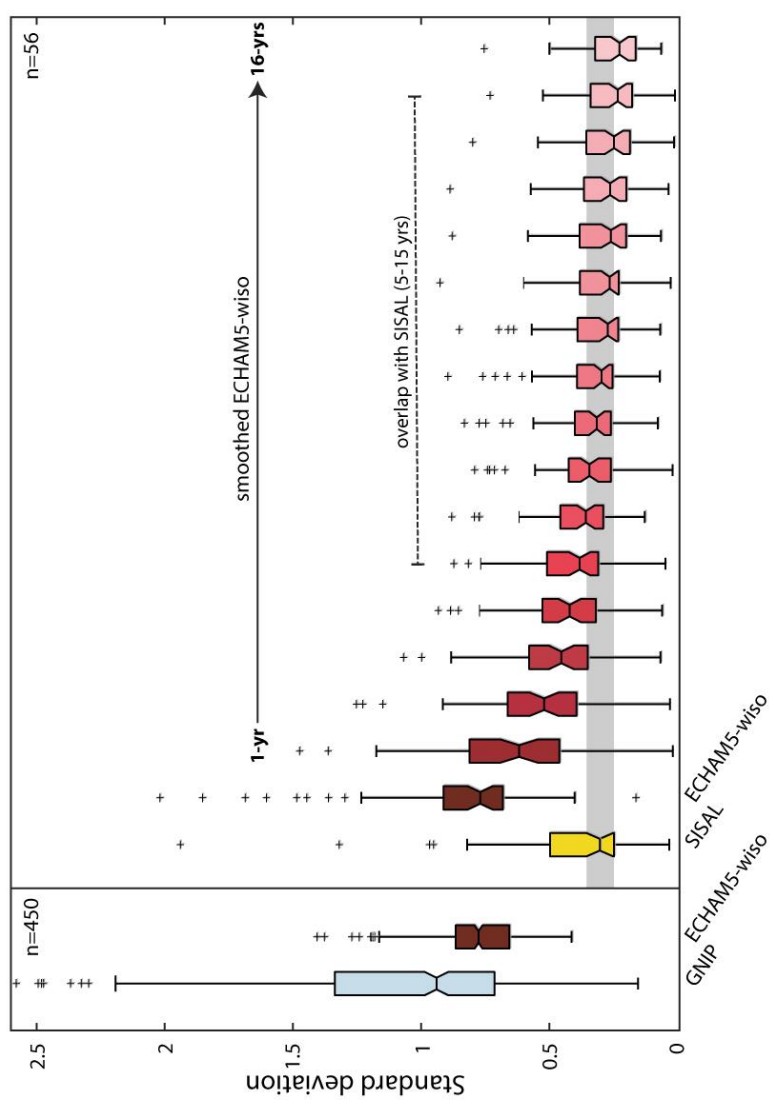





**Figure 5**

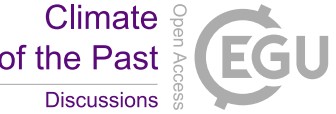



**Figure 6**

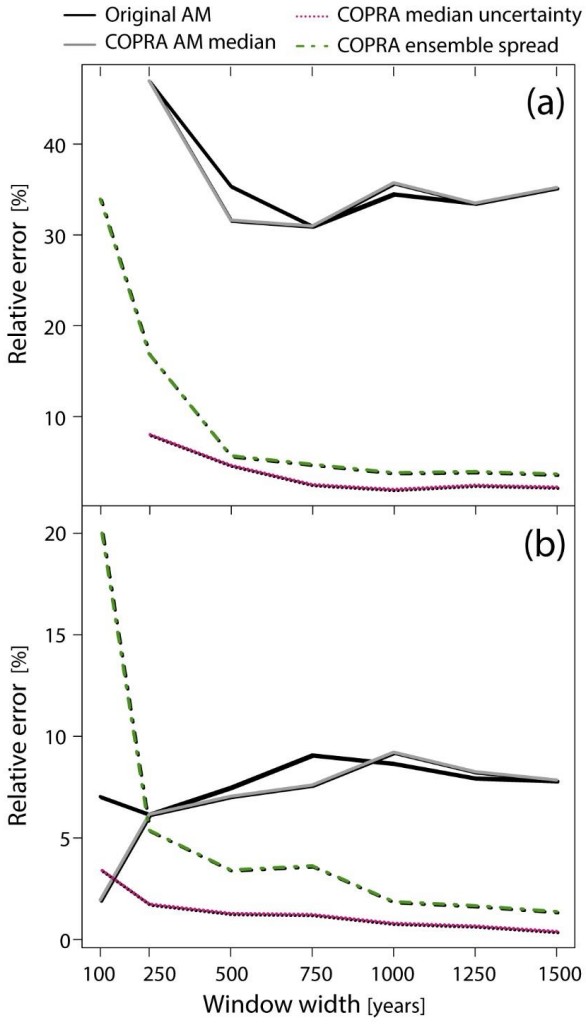



**Figure 7**

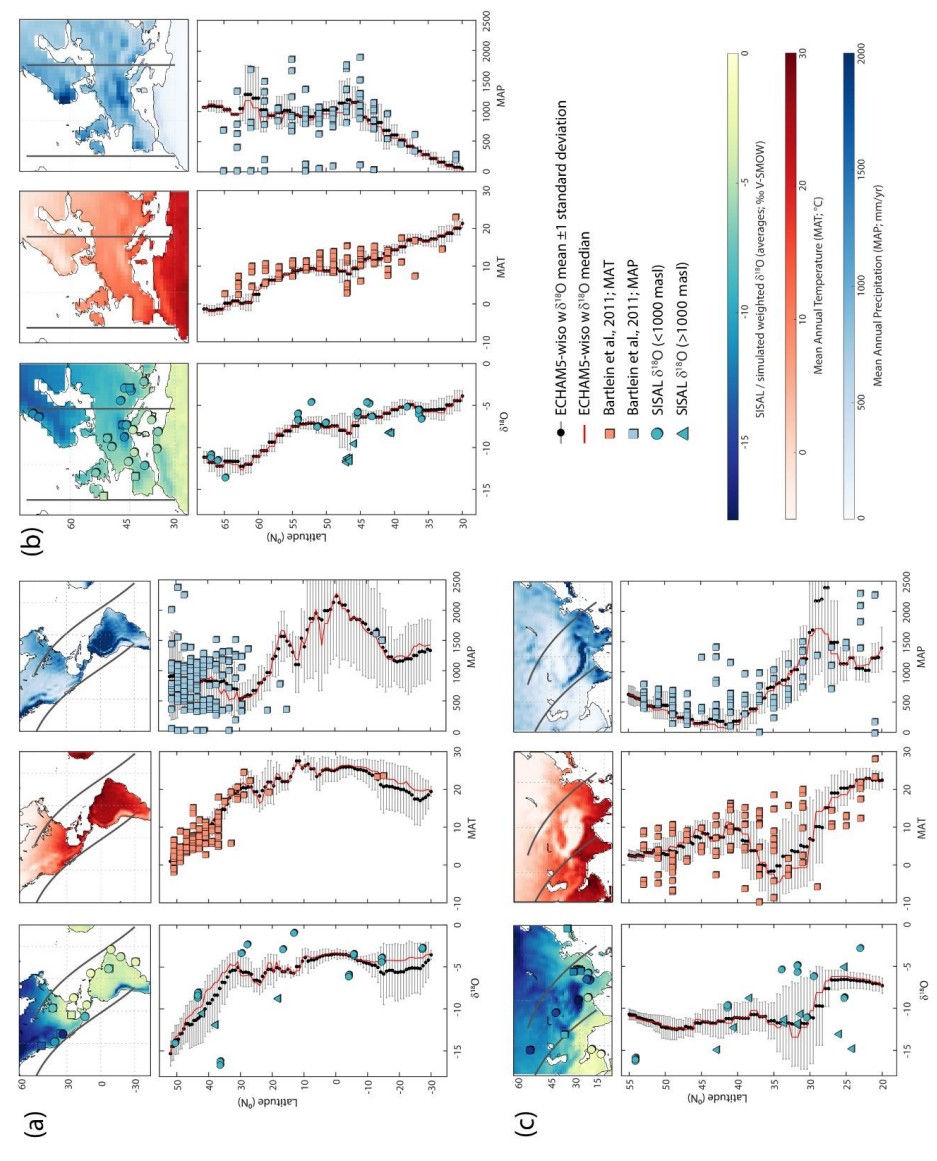





**Figure 8**

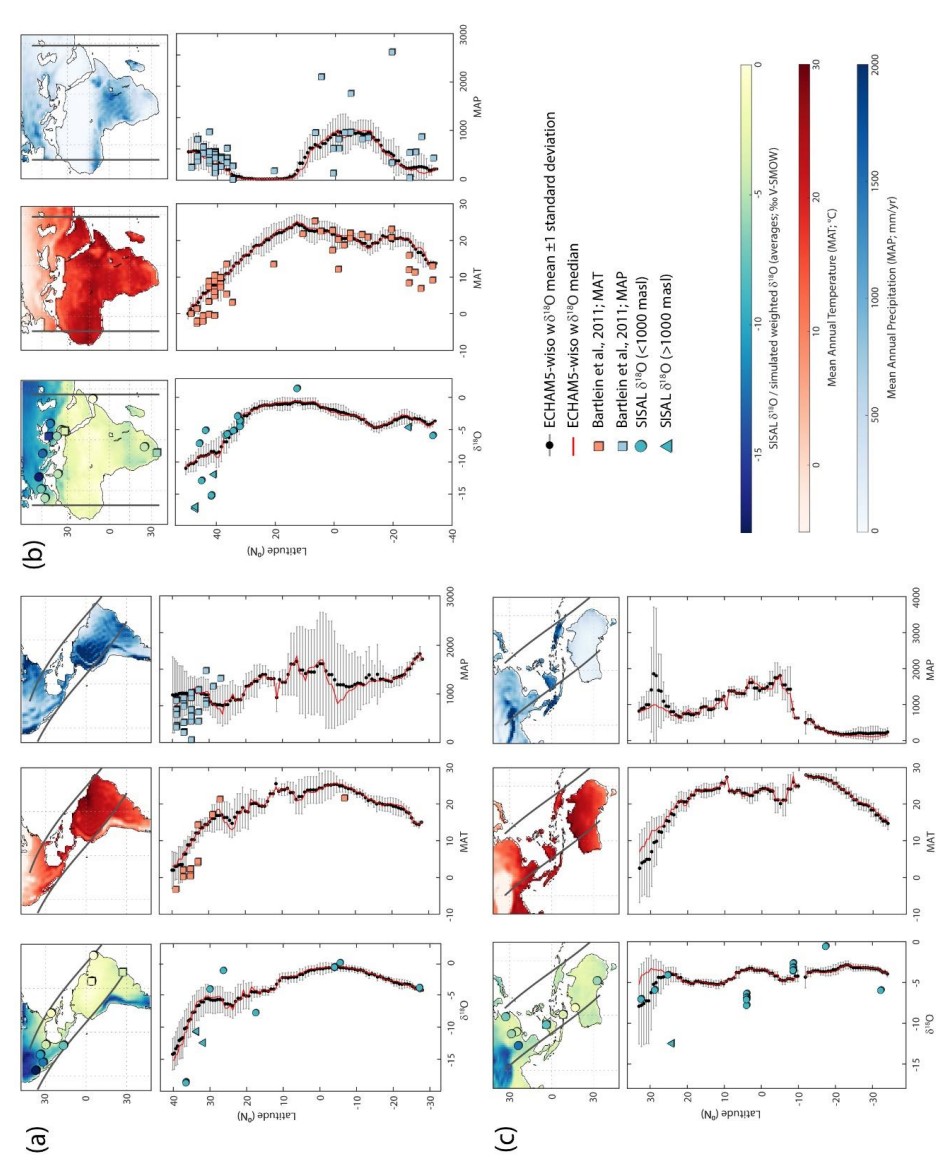






**Table 1:**

| Site name | Elev. | Lat. | Lon. | Entity name | Reference (s) |
|---|---|---|---|---|---|
| Arch cave | 660 | 50.55 | -127.07 | DM05-01 | Marshall et al. (2009) |
| Beatus cave | 875 | 46.38 | 7.49 | EXC3, EXC4 | Boch et al. (2011) |
| Bribin cave | 500 | -8.05 | 110.633 | JB2 | Hartmann et al. (2013) |
| Cesare Battisti cave | 1880 | 46.08 | 11.02 | CB25, CB39, CB47 | Johnston et al. (2018) |
| Chan Hol cave | -8.5 | 20.16 | -87.57 | CH-7 | Stinnesbeck et al. (2017) |
| Chen Ha cave | 550 | 16.6769 | -89.0925 | CH04-02 | Pollock et al. (2016) |
| Cold Water cave | 356 | 43.4678 | -91.975 | CWC-1s, CWC-2ss, CWC-3l | Denniston et al. (1999) |
| Devil's Icebox cave | 250 | 38.15 | -92.05 | DIB-1, DIB-2 | Denniston et al. (2007b) |
| Dongge cave | 680 | 25.2833 | 108.0833 | DA_2005, D4_2005 | Wang et al. (2005);Dykoski et al. (2005) |
| Dos Anas cave | 120 | 22.38 | -83.97 | CG | Fensterer et al. (2010);Fensterer et al. (2012) |
| El Condor cave | 860 | -5.93 | -77.3 | ELC_composite | Cheng et al. (2013) |
| Frasassi cave system - Grotta Grande del Vento | 257 | 43.4008 | 12.9619 | FR16 | Vanghi et al. (2018) |
| Goshute cave | 2000 | 40.0333 | -114.783 | GC_2, GC_3 | Denniston et al. (2007a) |
| Harrison's cave | 300 | 13.2 | -59.6 | HC-1 | Mangini et al. (2007);Mickler et al. (2004);Mickler et al. (2006) |
| Hoti cave | 800 | 23.0833 | 57.35 | H14 | Cheng et al. (2009);Fleitmann et al. (2003) |
| Jaraguá cave | 570 | -21.083 | -56.583 | JAR4, JAR1, JAR_composite | Novello et al. (2018);Novello et al. (2017) |
| Karaca cave | 1536 | 40.5443 | 39.4029 | K1 | Rowe et al. (2012) |
| Klaus Cramer cave | 1964 | 47.26 | 9.52 | KC1 | Boch et al. (2011) |
| KNI-51 cave | 100 | -15.18 | 128.37 | KNI-51-A1, KNI-51-P | Denniston et al. (2013) |
| Korallgrottan cave | 570 | 64.88 | 14.15 | K1 | Sundqvist et al. (2007) |
| Lianhua | 455 | 29.48 | 109.53 | A1 | Cosford et al. (2008a) |
| Lynds cave | 300 | -41.58 | 146.25 | Lynds_BCD | Xia et al. (2001) |
| Mawmluh cave | 1160 | 25.2622 | 91.8817 | MAW-0201 | Myers et al. (2015) |
| McLean's cave | 300 | 38.07 | -120.42 | ML2 | Oster et al. (2014) |
| Minnetonka cave | 2347 | 56.5833 | -119.65 | MC08-1 | Lundeen et al. (2013) |
| Moondyne cave | 100 | -34.27 | 115.08 | MND-S1 | Fischer and Treble (2008);Nagra et al. (2017) Treble et al. (2003);Treble et al. (2005) |

| Paraiso cave | 60 | -4.0667 | -55.45 | Paraiso composite | Wang et al. (2017) |
|---|---|---|---|---|---|
| Peqiin cave | 650 | 32.58 | 35.19 | PEK_composite, PEK 6, PEK 9, PEK 10 | Bar-Matthews et al. (2003) |
| Piani Eterni karst system | 1893 | 46.16 | 11.99 | MN1, GG1, IS1 | Columbu et al. (2018) |
| Poleva cave | 390 | 44.7144 | 21.7469 | PP10 | Constantin et al. (2007) |
| São Bernardo cave | 631 | -13.81 | -46.35 | SBE3 | Novello et al. (2018) |
| São Matheus cave | 631 | -13.81 | -46.35 | SMT5 | Novello et al. (2018) |
| Shatuca cave | 1960 | -5.7 | -77.9 | Sha-2, Sha-3, Sha-composite | Bustamante et al. (2016) |
| Sofular cave | 440 | 41.42 | 31.93 | So-17A, So-2 | Badertscher et al. (2011);Fleitmann et al. (2009) Göktürk et al. (2011) |
| Soylegrotta cave | 280 | 66 | 14 | SG93 | Lauritzen and Lundberg (1999) |
| Tangga cave | 600 | -0.36 | 100.76 | TA12-2 | Wurtzel et al. (2018) |
| Uluu-Too cave | 1490 | 40.4 | 72.35 | Uluu2 | Wolff et al. (2017) |
| White moon cave | 170 | 37 | -122.183 | WMC1 | Oster et al. (2017) |
| Xiangshui cave | 380 | 25.25 | 110.92 | X3 | Cosford et al. (2008b) |
| Xibalba cave | 350 | 16.5 | -89 | GU-Xi-1 | Winter et al. (2015) |
| Yaoba Don cave | 420 | 28.8 | 109.83 | YB | Cosford et al. (2008b) |

**Table 2:**

| Time period | Number of speleothems (entities) and cave sites in both periods |
|---|---|
| Modern (1961–1990 CE) | 58 entities (47 sites) |
| PI (1835–1865 CE) | 62 entities (51 sites) |
| Extended PI (1850–1990 CE) | 87 entities (69 sites) |
| MH and PI | 18 entities (17 sites) |
| MH and extended PI | 34 entities (29 sites) |
| MH and Last Millennium (LM, 850–1850 CE) | 48 entities (38 sites) |
| LGM and PI | 5 entities (5 sites) |
| LGM and extended PI | 11 entities (10 sites) |
| LGM and Last Millennium (LM, 850–1850 CE) | 12 entities (10 sites) |
| LGM and MH | 20 entities (16 sites) |