# Peer review of "Evaluating model outputs using integrated global speleothem records of climate change since the last glacial"

_Climate of the Past, 2019_

## Short Comment (SC1) · 7 Mar 2019

Dear All,

The V-PDB to V-SMOW conversion equation at line 212 is wrong. The correct equation is d18O(V-SMOW)=1.03091*d18O(V-PDB)+30.91.

The difference between the two equations is 1.2 to 1.8 ‰ in the range of 0 to -10 ‰ V-PDB.

Attila Demény

---

## Referee Comment (RC1) · Anonymous Referee #1 · 25 Mar 2019

The paper compares isotopic simulations with ECHAM5-wiso for present day, last glacial maximum (LGM) and mid-holocene (MH) to speleothem records archived in the new SISAL database. They propose recommendations for an optimal model-data comparison, which can be useful for future such comparisons.

The paper is well-written, although it could sometimes be made more concise. The figures are of good quality.

Besides minor comments listed below, I have one major comment: the authors argue that it is useful for model evaluation to look at spatial patterns of absolute $\delta^{18}O$ for past climates (LGM, MH) rather than just looking at anomaly maps, in contrast with many

previous studies. However, the argument is not convincing and the examples given argue rather for the contrary. I would suggest to remove this sub-section and remove this recommendation.

**1 Major comment: what is the added value of looking at spatial patterns for past climates compared to looking at them for present-day?**

The authors argue that it is useful for model evaluation to look at spatial patterns of absolute $\delta^{18}O$ for past climates (LGM, MH) rather than just looking at anomaly maps, in contrast with many previous studies. This allows to have more sites for model-data comparison. However, what is the added value of looking at spatial patterns for past climates compared to looking at them for present-day? For present-day, spatial patterns would be the same to first order. At present, there are so many more sites available directly samplig precipitation (GNIP), so why bother with speleothem records for past climates?

- Figures 7 and 8 show the spatial patterns of observed and simulated $\delta^{18}O$ for LGM and MH. The sub-figure a representing North and South America are common to both figures, and they actually show very similar patterns. The same figure for present day would also show very similar patterns. This is because $\delta^{18}O$, temperature or precipitation changes between LGM, MH and present-day are much smaller than the magnitude of spatial variations along a transect covering such a wide range of latitudes. So these figures support my skepticism about the relevance of spatial patterns in absolute values for past climates.
  Figures 7b,c and 8b,c do not represent the same regions. But I'm sure that the maps for present-day would look very similar.

- The corresponding text bears several slips of the pen and/or interpretation errors, that probably reflect that writing this sub-section was not confortable:

[Figure]

– l 408: "$\delta^{18}O$ changes" should be replaced by "$\delta^{18}O$ patterns": the authors write "changes" because this is really what is interesting to look at, but actually the figures do not show it.

– l 412: "underestimates changes in precipitation": again, we cannot see changes from present-day to MH on this figure.

What you want to plot depends on the science question. If the science question is what controls spatial patterns in absolute values, then it's better to focus on present-day values; past climates do not provide much added value. But if the science question is what controls the changes at paleo-climatic time scales, then it is necessary to look at anomalies between 2 climatic states.

-> So I recommend to remove section 3.4, or replace it by an analysis of spatial patterns of anomalies, and to modify accordingly the abstract, protocol and conclusions.

**2 Detailed comments**

- l 157-158: I don't underestand what this means. Where is the control simulations included in the LGM-MH difference?

- l 209: remove "non-equilibrium of"

- l 216: add a dot.

- remove "with data ... baseline".

- l 234-235: already said, remove.

- 246: define *pchip*

- l 295: clarify the rationale. Why can't there be a sampling bias in temperate regions towards the PMIP periods?

- l 296: clarify this sentence. What does "even at a global level" mean?

- l 300-302: can you explain briefly why higher latitude speleothems are more depleted than OIPC and low latitude speleothems are more enriched? l 305: "cave specific factors" cannot explain why you have such systematic differences common to wide regions.

- l 317-319: should the reader conclude that ECHAM underestimates the inter-annual variability? If so, please state this clearly. Has such a bias already been described in a previous paper, for ECHAM or for other models? Explain briefly what could be the reason for this underestimate.

- move "processes" before "within"

- l 325-328: this has already been said just above.

- l 359: replace "anomalies" by "MH values"?

- l 422: remove "utilising"

- l 440-447: this issue was not previously discussed. Add some quantification, or a map, showing what error we would make if we use only the fractionation factor for calcite?

- l 473: remove "on a global basis". Or clarify what you mean. Even at a specific cave, if the speleothem acts as a low pass filter, time scales shorter than "quasi-decadal" cannot be studied.

- l 475-476: clarify. Do you mean that the model underestimates $\delta^{18}O$ changes? l 476-477: clarify.

- l 489: "constraining structural error on the model side": what do you mean? There are plenty of sources of errors in the model: errors on forcings, missing processes in parameterization package, tunable parameters, coarse resolution... Which one are you refering to? "true uncertainties": beware that errors are not the same as uncertainties. Anyway, in this paragraph, I suggest to focus only on uncertainties on the observation side, because this is what is useful to evaluate models. The question of quantifying uncertainties in models is set differently and is beyond the scope of this paper.

———————————————————

---

## Author Comment (AC1) · 3 Apr 2019

Dear Attila Démeny,

thanks for noticing the mistake in the equation used to convert V-PDB values to the V-SMOW scale.

We have now re-done the calculations with the correct equation and can confidently say that the conclusions presented in our manuscript do not change. The revised V-SMOW values are at a constant offset of ∼1 permil from the ones initially presented and the impact of this correction is only noticeable on the intercept of the regressions

presented in Figure 3 and Supplementary Figures 1 and 2. This correction has no noticeable impact on the spatial transects presented in Figures 7 and 8.

We will revise the text and figures with the corrected values.

Best regards,

Laia Comas Bru

––––––––––––––––––––––––––––––––

---

## Referee Comment (RC2) · Anonymous Referee #2 · 6 Apr 2019

Though focusing most of my research interests on paleoceanography and spending most of my time on producing proxy records in paleoceanography, I also have great interests for comparison of paleoenvironmental proxy records with modelling results. Speleothem records have great significance in improving our understanding on hydrological cycles on timescales from orbital to millennial to centennial. Because speleothem records have much higher time resolution than the lake and marine records and could be precisely absolutely dated, they could serve as excellent target records for testing the simulation results of the Earth System Model. Therefore, they are very useful in refining our ESM model and thus promote our level of predicting future climate change.

[Figure]

However, changes of speleothem records have temporal and spatial difference. Therefore, a good speleothem dataset which integrates different regional records and has global significance is the key to the final success. The updated SISAL database (SISALv1b) (SISAL, Speleothem Isotopes Synthesis and Analysis, an international working group under the auspices of the PAGES project) is such a dataset based on my evaluation.

Additionally, to achieve such a success on model-data comparison, a reliable Earth System Model is another key because different Earth System Models also produce unexpected biases. The ECHAM5-wiso used for their simulation is such a reliable model. It is an isotope-enabled atmosphere GCM, of which the consideration of the water cycle is very good, which contains formulations for evapotranspiration of terrestrial water, evaporation of ocean water, and the formation of large-scale and convective clouds. The achievements using this model for climate and paleoclimate research are productive and of high reputation. The most advantage of this model is the high resolution. As the authors explains, all the ECHAM5-wiso simulations were run at T106 horizontal grid resolution (approx. 1.1°x1.1°) with 31 vertical levels.

My overall evaluation on their data-model comparison is the same as the authors stated in this manuscript that the simulations succeeded in catching the 1st order trend of records, which could be seen in Figures 5, 7 and 8.

This manuscript is well written in language though the structure could be much simpler so as to make the reading easier for most readers. For example, they could move the contents related to methods to the supplementary and focus mainly on the results and discussion. This can make the reading more consistent.

Minor issues. Are the control runs for MH and LGM different? Probably I don't catch the points. In my understanding, they should be the same which is the base for probing the climatic significance of the difference between the MH and LGM simulation experiments.

The simulation results of the MH seem to be better than that of the LGM. Could they explain more on this? For example, they use the protocol of PIMP3 for the LGM modelling, and their SST forcing is based on the results of a full transient experiment. More clarification on why they take such steps will make this manuscript more convinced.

I noticed that the other reviewers gave many more professional comments. I think that this paper has great potential for publication in CP after minor revisions.

---

## Referee Comment (RC3) · Anonymous Referee #3 · 18 Apr 2019

The authors used different approaches to compare the speleothem records from the SISAL database with the simulated results of the ECHAM5-wiso model for present-day, MH and LGM. Based on their analyses, they propose a protocol for using speleothem isotopic data for model evaluation. The paper is well written and the analyses could be interesting for researchers working in related field. However, it seems to me that the paper could be improved by adding more in-depth discussions/analyses.

I don't see very well what is the advancement made by this study as compared to the traditional approach for comparing the speleothem records with models. Maybe the authors should stress more why their approaches are better and what new can been

discovered by their approaches that can not be done by traditional approach.

It is not very clear to me what is the final goal of the data-model comparison and what can be improved or learned after all the analyses. If the comparison is good, can we assume that the temperature and precipitation simulated by the model are correct and what is the uncertainty? What might be the reasons for the similarities and differences between model results and speleothem data? Can the results help to improve the model and/or experiment design and how?

The major uncertainties and biases of the ECHAM5-wiso model in simulating present-day and past climates and the experiment design of the MH and LGM simulations, the reliability of the SST and sea ice simulated by the CCSM3 and their potential influence on the data-model comparison should be discussed.

The simulations for MH and LGM are only 12 and 22 years. Are they long enough to allow the climate at different speleothem location reaching equilibrium? What is the initial state of these simulations? What might be the influence of using fixed ocean condition?

---

## Author Comment (AC2) · 25 Apr 2019

**Reply to Anonymous Referee 2**

Though focusing most of my research interests on paleoceanography and spending most of my time on producing proxy records in paleoceanography, I also have great interests for comparison of paleoenvironmental proxy records with modelling results. Speleothem records have great significance in improving our understanding on hydrological cycles on timescales from orbital to millennial to centennial. Because speleothem records have much higher time resolution than the lake and marine

records and could be precisely absolutely dated, they could serve as excellent target records for testing the simulation results of the Earth System Model. Therefore, they are very useful in refining our ESM model and thus promote our level of predicting future climate change. However, changes of speleothem records have temporal and spatial difference. Therefore, a good speleothem dataset which integrates different regional records and has global significance is the key to the final success. The updated SISAL database (SISALv1b) (SISAL, Speleothem Isotopes Synthesis and Analysis, an international working group under the auspices of the PAGES project) is such a dataset based on my evaluation. Additionally, to achieve such a success on model-data comparison, a reliable Earth System Model is another key because different Earth System Models also produce unexpected biases. The ECHAM5-wiso used for their simulation is such a reliable model. It is an isotope-enabled atmosphere GCM, of which the consideration of the water cycle is very good, which contains formulations for evapotranspiration of terrestrial water, evaporation of ocean water, and the formation of large-scale and convective clouds. The achievements using this model for climate and paleoclimate research are productive and of high reputation. The most advantage of this model is the high resolution. As the authors explains, all the ECHAM5-wiso simulations were run at T106 horizontal grid resolution (approx. 1.1âŮęx1.1âŮę) with 31 vertical levels. My overall evaluation on their data-model comparison is the same as the authors stated in this manuscript that the simulations succeeded in catching the 1st order trend of records, which could be seen in Figures 5, 7 and 8.This manuscript is well written in language though the structure could be much simpler so as to make the reading easier for most readers. For example, they could move the contents related to methods to the supplementary and focus mainly on the results and discussion. This can make the reading more consistent.

*We thank the reviewer for their comments. We try to clarify what we are doing in response to these comments: the reviewer comments are in black bold and our explanations in blue italics.*

**Minor issues.**

Are the control runs for MH and LGM different? Probably I don't catch the points. In my understanding, they should be the same which is the base for probing the climatic significance of the difference between the MH and LGM simulation experiments.

*Yes, the control simulations differ slightly in the prescribed monthly mean sea surface temperatures and sea ice cover data. Details of the different simulation setups are given in Wackerbarth et al. (2012) and Werner et al. (2018). However, both control simulations are more similar than the differences between MH-PI and LGM-PI. These different PIs have been taken into account in Figure 6 (anomaly maps) and we will add a line in the text clarifying how this has been done (as suggested by Referee 1).*

The simulation results of the MH seem to be better than that of the LGM. Could they explain more on this? For example, they use the protocol of PIMP3 for the LGM modelling, and their SST forcing is based on the results of a full transient experiment. More clarification on why they take such steps will make this manuscript more convinced.

*We cannot say that the MH simulation results fit better to the speleothem data compared to the LGM simulation due to the limited number of speleothem records available. However, we will add in the manuscript that for the MH simulation, we have chosen a simulation which fitted best to European stalagmite data. For the details on this, we will refer the reader to Wackerbarth et al., 2012, where three different ECHAM5-wiso MH simulations are compared.*

---

## Author Comment (AC3) · 25 Apr 2019

**Reply to Anonymous Referee 3**

The authors used different approaches to compare the speleothem records from the SISAL database with the simulated results of the ECHAM5-wiso model for present-day, MH and LGM. Based on their analyses, they propose a protocol for using speleothem isotopic data for model evaluation. The paper is well written and the analyses could be interesting for researchers working in related field. However, it seems to me that the paper could be improved by adding more in-depth discussions/analyses.

[Figure]

*We thank the reviewer for their comments, but we feel they may not have fully understood the purpose of this paper. We try to clarify what we are doing in response to these comments: the reviewer comments are in normal script in black, our explanations in blue italics and additional text is given in normal script in blue.*

I don't see very well what is the advancement made by this study as compared to the traditional approach for comparing the speleothem records with models. Maybe the authors should stress more why their approaches are better and what new can been discovered by their approaches that cannot be done by traditional approach.

*Data-model comparisons using speleothem data are comparatively new, and have tended to focus on validation of new versions of isotope-enabled models. These comparisons have often overlooked important characteristics of, and/or uncertainties associated with, the speleothem records (see discussion lines 90-99). There is no agreed protocol for using speleothem data for model evaluation. The purpose of our paper was to identify issues that could affect data-model comparisons, drawing on the new SISAL database that has been explicitly constructed to facilitate such comparisons and the expertise of the speleothem experts who constructed this database. Thus we are not claiming that our approach is different from or better than a "traditional" approach – we are simply making it clear how speleothem data should and could be used. We can make this clearer by amplifying our description of the purpose of the paper (108-110) as follows:*

In this paper, we examine a number of issues that need to be addressed in order to use speleothem data, most especially data from the SISAL database, for model evaluation in the palaeoclimate context and make recommendations about robust

approaches that should be used for model evaluation in CMIP6-PMIP4. We focus particularly on interpretation issues that could be overlooked in using the speleothem records and we show the strengths and limitations of different comparison techniques.

It is not very clear to me what is the final goal of the data-model comparison and what can be improved or learned after all the analyses. If the comparison is good, can we assume that the temperature and precipitation simulated by the model are correct and what is the uncertainty? What might be the reasons for the similarities and differences between model results and speleothem data? Can the results help to improve the model and/or experiment design and how?

*As we explain in the introduction (lines 45-54), model evaluation using palaeoclimate data provides an out-of-sample test of model performance and is one component of the Palaeoclimate Modelling Intercomparison Project. Such evaluations help to provide confidence in the projections of future climates. Speleothems are a relatively new source of information for such evaluations and the purpose of our paper is to provide a robust framework to make such evaluations. We do not want to distract from this goal by discussing the generic purposes of data-model comparison in the Introduction to the paper, but we could certainly add a concluding paragraph discussing what can be learnt from such data-model comparisons as follows:*

Comparisons with speleothem data can be seen as a complement to model evaluation using other types of palaeoenvironmental data and palaeoclimatic reconstructions (see e.g. MARGO Project Members, 2009; Harrison et al., 2014). They can be considered particularly useful because they provide insights into how well state-of-the-art models reproduce the hydrological cycle and atmospheric circulation patterns. The ability to reproduce past observations provides additional confidence in the ability of climate models to simulate large climate changes, such as those expected by the end of the
21st century (Braconnot et al., 2012; Schmidt et al., 2014). However, mismatches between model simulations and palaeo-observations are also useful because they can help to pinpoint issues that may need to be addressed in developing improved models or in better experimental protocols (Kageyama et al., 2018), providing that these mismatches do not arise because of misunderstanding or misinterpretation of the observations themselves. By providing a protocol for using speleothem data for data-model comparisons that accounts for uncertainties in the observations, we anticipate that at least such causes of data-model mismatches will be minimized.

The major uncertainties and biases of the ECHAM5-wiso model in simulating present day and past climates and the experiment design of the MH and LGM simulations, the reliability of the SST and sea ice simulated by the CCSM3 and their potential influence on the data-model comparison should be discussed.

*We use outputs from the ECHAM-wiso model in order to illustrate potential approaches to data-model comparison. Our goal here is not to provide an in-depth evaluation of the quality of these simulations. The performance of the ECHAM-wiso model under modern day conditions has been extensively analysed (see e.g. Werner et al., 2011; Wackerbarth et al., 2012) and the MH and LGM simulations have also been published and discussed (Wackerbarth et al., 2012; Werner et al., 2018). In order to make it clear that our use of the model is illustrative, we will modify the final section of the introduction (line 113 onwards) to read:*

We use an updated version of the SISAL database (SISALv1b: Atsawawaranunt et al., 2019) and simulations made with the ECHAM5-wiso isotope-enabled atmospheric circulation model (Werner et al., 2011) to explore the various issues in making data-model comparisons. The goal is not to evaluate the ECHAM5-wiso simulations but rather to use them to illustrate generic issues in data-model comparison with

speleothem isotopic data.

The simulations for MH and LGM are only 12 and 22 years. Are they long enough to allow the climate at different speleothem location reaching equilibrium? What is the initial state of these simulations? What might be the influence of using fixed ocean condition?

*As we explain in the methods section (lines 128-154), these are atmosphere-only simulations forced with sea-surface temperatures and sea-ice cover from a pre-existing transient simulation. Thus, there is no spin-up necessary and the issue of equilibrium is irrelevant. If the purpose of this paper were to use the model simulations to explain speleothem records, then the lack of ocean coupling would mean that the simulations would be unsuitable for evaluating the degree to which long-term (multi-decadal) variability in the speleothem isotope record reflected internal unforced variability. But as our purpose in using the experiments is illustrative, then the short length of the simulations is not important. We hope that the modification to the introduction suggested above will help clarify the purpose of this paper.*

---

## Author Comment (AC4) · 25 Apr 2019

**Reply to Anonymous Referee 1**

The paper compares isotopic simulations with ECHAM5-wiso for present day, last glacial maximum (LGM) and mid-holocene (MH) to speleothem records archived in the new SISAL database. They propose recommendations for an optimal model-data comparison, which can be useful for future such comparisons.

[Figure]

The paper is well-written, although it could sometimes be made more concise. The figures are of good quality. Besides minor comments listed below, I have one major comment: the authors argue that it is useful for model evaluation to look at spatial patterns of absolute $\delta^{18}$O for past climates (LGM, MH) rather than just looking at anomaly maps, in contrast with many previous studies. However, the argument is not convincing and the examples given argue rather for the contrary. I would suggest to remove this sub-section and remove this recommendation.

*We thank the reviewer for their comments. We try to clarify what we are doing in response to these comments: the reviewer comments are in normal script, our explanations in blue italics and the rest of the text in blue normal script.*

**1 Major comment: what is the added value of looking at spatial patterns for past climates compared to looking at them for present-day?**

The authors argue that it is useful for model evaluation to look at spatial patterns of absolute $\delta^{18}$O for past climates (LGM, MH) rather than just looking at anomaly maps, in contrast with many previous studies. This allows to have more sites for model data comparison. However, what is the added value of looking at spatial patterns for past climates compared to looking at them for present-day? For present-day, spatial patterns would be the same to first order. At present, there are so many more sites available directly sampling precipitation (GNIP), so why bother with speleothem records for past climates?

Figures 7 and 8 show the spatial patterns of observed and simulated $\delta^{18}$O for LGM and MH. The sub-figure a representing North and South America are common to both figures, and they actually show very similar patterns. The same figure for present

day would also show very similar patterns. This is because $\delta^{18}$O, temperature or precipitation changes between LGM, MH and present-day are much smaller than the magnitude of spatial variations along a transect covering such a wide range of latitudes. So these figures support my skepticism about the relevance of spatial patterns in absolute values for past climates. Figures 7b,c and 8b,c do not represent the same regions. But I'm sure that the maps for present-day would look very similar.

The corresponding text bears several slips of the pen and/or interpretation errors, that probably reflect that writing this sub-section was not comfortable:

– l 408: "$\delta^{18}$O changes" should be replaced by "$\delta^{18}$O patterns": the authors write "changes" because this is really what is interesting to look at, but actually the figures do not show it.

– l 412: "underestimates changes in precipitation": again, we cannot see changes from present-day to MH on this figure.

What you want to plot depends on the science question. If the science question is what controls spatial patterns in absolute values, then it is better to focus on present day values; past climates do not provide much added value. But if the science question is what controls the changes at paleo-climatic time scales, then it is necessary to look at anomalies between 2 climatic states.

So I recommend to remove section 3.4, or replace it by an analysis of spatial patterns of anomalies, and to modify accordingly the abstract, protocol and conclusions.

*We agree that we need to make the case that using palaeodata over different transects is valuable for data-model comparisons in addition to looking at anomalies between two climatic states. We expect the gradients shown in Figures 7 and 8 to change over time due to the large ice-sheets during the LGM and the different insolation patterns at different latitudes during the MH. In some cases, the model will be able to simulate those gradients/patterns and in some other cases, the model will not – and this is valuable information.*

*We have therefore decided to modify the current Figures 7 and 8 to better showcase the changes that occur in the transects between the modern (1958-2013 CE), the MH and the LGM. As a result of this, we will remove the pollen-based reconstructions - which are not the focus of this paper - to only focus on focus on isotope data.*

*The two examples enclosed as Review Figures 1 and 2 will be part of these new figures. In the case of the Asian transect, we observe a fundamental change in the latitudinal gradient across time periods and in particular during the MH. The SE-NW gradient in the data is clearly not reproduced by the model, which systematically simulates more isotopically enriched precipitation between 20-35 N. On the other hand, the Asian-Oceania transects are fairly similar across time periods but the clear offset during the MH between the data and the model supports the fact that the model underestimates the intensification of the hydrological cycle during this period. We hope the reviewer will agree that this observation is clearer in the spatial transects than in the traditional anomaly plots shown in Figure 6.*

*We will rewrite section 3.4 to reflect the changes in the figures.*

**2 Detailed comments**

L157-158: I do not understand what this means. Where is the control simulations included in the LGM-MH difference?

*Two different control simulations are available for the lgm and 6ka experiments and to avoid any potential offset between these control simulations being incorporated in the LGM-MH anomaly map (Figure 6c), we have calculated the LGM-MH anomalies as (lgm-PI$_{lgm}$) − (6ka-PI$_{6ka}$) instead of doing directly lgm-6ka. We will clarify this in the text with the following sentence:*

We also calculated the anomaly between the LGM and MH (LGM-MH), taking account of the difference between their control simulations in the following way: (lgm-PI$_{lgm}$) − (6ka-PI$_{6ka}$).

L209: remove "non-equilibrium of"

*We will do this.*

L216: add a dot.

*We will do this*

L223: remove "with data ... baseline".

*Removing "with data ... baseline" as suggested by the reviewer obscures what palaeo-climate simulations are being compared to: the data. We will clarify this sentence by writing:*

Data-model comparisons are generally made by comparing (1) anomalies between a control period and a palaeoclimate simulation with (2) data anomalies with respect to a modern baseline.

L234-235: already said, remove.

*We agree with the reviewer that this has already been mentioned in L156-158 and the text will be removed.*

246: define pchip

*We will add the definition of pchip (piecewise cubic hermite interpolation) in the text.*

L295: clarify the rationale. Why can't there be a sampling bias in temperate regions towards the PMIP periods?

*The previous sentence already suggested that the deviations could result from a sampling bias. However, we will rephrase this sentence to clarify this point:*

These deviations could arise from sampling biases but it is unlikely that human sampling bias would be different in the tropics than in temperate regions. Differences between the deviation curves of both regions curves at least for the last 130ka (Figure

2b,c) suggest increased climate variability in the extra-tropics leads to increased deviation from expected stalagmite growth.

L296: clarify this sentence. What does even "at a global level" mean?

*We will rephrase this sentence to:*

Thus, the speleothem data provide a first-order assessment of climate changes on orbital time scales globally.

L300-302: can you explain briefly why higher latitude speleothems are more depleted than OIPC and low latitude speleothems are more enriched?

*Here, we were interpreting the map patterns rather than doing a robust assessment of the observed trends. We believe that these patterns will probably not be significant and, as we do not see any reason why this may be the case, we will take this sentence out.*

L305:"cave specific factors" cannot explain why you have such systematic differences common to wide regions.

*We agree with the reviewer and refer to our answer above.*

L317-319: should the reader conclude that ECHAM underestimates the interannual variability? If so, please state this clearly. Has such a bias already been described in a previous paper, for ECHAM or for other models? Explain briefly what could be the

reason for this underestimate.

*According to the AR5 IPCC report, although there has been substantial progress between CMIP3 and CMIP5 models in their ability to simulate precipitation extremes, there is a tendency to underestimate the sensitivity of extreme precipitation to temperature variability or trends and, in turn, its inter-annual variability. In particular, ECHAM5 underestimates this inter-annual variability in regions with prevalence of convective precipitation (i.e., the tropics), as well as in extratropical regions during the summer (e.g., in southern Europe) (see Eden et al., 2012). This is due to the fact that formation of convective precipitation acts on small scales and has a large random component, even for a given large-scale atmospheric state. In another study (Butzin et al., 2014), the authors found that for three Siberian GNIP stations $\delta^{18}O$-variability was also underestimated. We will modify the manuscript to incorporate this information.*

References not already listed in the manuscript:

Eden, J. M., Widmann, M., Grawe, D. and Rast, S.: Skill, Correction, and Downscaling of GCM-Simulated Precipitation, J. Climate, 25(11), 3970–3984, doi:10.1175/JCLI-D-11-00254.1, 2012.

Flato, G., Marotzke, J., and others:Evaluation of climate models, in: Climate Change 2013 – The Physical Science Basis: Working Group I Contribution to the Fifth Assessment Report of the Intergovernmental Panel on Climate Change, edited by: Change, I. P. C. C., Cambridge University Press, Cambridge, 741-866, 2013.

L321: move "processes" before "within"

[Figure]

*We will revise this sentence to read:*

*(...) reflecting the impact of karst and in-cave processes that effectively act as a low-pass filter (...)*

L325-328: this has already been said just above.

*We agree with the reviewer. Both sentences will be merged to avoid duplication.*

L359: replace "anomalies" by "MH values"?

*We agree with the reviewer and we will make the suggested change in the text.*

L422: remove "utilising"

*We will rephrase this sentence to:*

Our analyses illustrate a number of possible approaches for using speleothem isotopic data for model evaluation.

L440-447: this issue was not previously discussed. Add some quantification, or a map, showing what error we would make if we use only the fractionation factor for calcite?

*While the impact of using one of another fractionation equation is minimal ( i.e. smaller than the measurement uncertainty) for sites with MAT > 27.3C, the added*

*uncertainty is noticeable at sites with lower MAT. Changes of MAT across time periods could exacerbate these differences for individual sites thus making the calculated entity-based anomalies inaccurate. As requested by the reviewer, we will add a new supplementary figure (enclosed as Review Figure 3) to showcase that using the appropriate correction according to the speleothem's mineralogy is important. We will also revise the text accordingly.*

L473: remove "on a global basis". Or clarify what you mean. Even at a specific cave, if the speleothem acts as a low pass filter, time scales shorter than "quasidecadal" cannot be studied.

*The temporal smoothing inflicted by the karst processes varies from site to site. There are sites where the transmission from the surface to the cave happens rapidly and as a result speleothems preserve yearly or even sub-annual variability. However, in using the database to construct regional signals, there will always be some sites that have a high temporal smoothing, and therefore you cannot use them for timescales shorter than quasi-decadal (as seen in Figure 4). We will revise the text to clarify this point.*

L475-476: clarify. Do you mean that the model underestimates $\delta^{18}$O changes?

*Yes, the model underestimates the amplitude of $\delta^{18}O$ changes as recorded in the speleothems. For details we refer the reviewer to our answers on their comment on L317-319. We will rephrase this sentence to:*

Using the traditional anomaly approach to data-model comparisons, consistency between the sign of observed and simulated changes in both the MH and the LGM exists. However, the ECHAM5-wiso model underestimates $\delta^{18}$O compare to the

speleothems.

L476-477: clarify.

*We agree that this sentence is too broad and general and have decided to delete it.*

L489: "constraining structural error on the model side": what do you mean? There are plenty of sources of errors in the model: errors on forcings, missing processes in parameterization package, tunable parameters, coarse resolution... Which one are you refering to? "true uncertainties": beware that errors are not the same as uncertainties. Anyway, in this paragraph, I suggest to focus only on uncertainties on the observation side, because this is what is useful to evaluate models. The question of quantifying uncertainties in models is set differently and is beyond the scope of this paper.

*We will rephrase this sentence to:*

There are many reasons why climate models do not simulate observed climate changes, including lack of key forcings, missing processes, structural errors, coarse resolution, etc. However, in this paper we focused on potential uncertainties on the speleothem data. Site-specific controls...
* * *
**Figure captions:**

**Review Figure 1:**Modern, Mid-Holocene (MH) and Last Glacial Maximum (LGM) transects for Asia. Maps at the top of each panel show the simulated $\delta^{18}Op$ from

ECHAM5-wiso. All transects show absolute $\delta^{18}$O values. In the maps, filled circles are SISAL $\delta^{18}$O averages for entities that cover both the MH and the modern reference period. Filled squares are SISAL entities that do not have a corresponding modern. Bottom plots of each panel show the simulated data extracted for each transect: black circles and whiskers are mean $\pm 2$ standard deviation of the data extracted along longitudinal sections in between the two great circle lines shown in dashed grey lines in the top maps. The red line is the median of the extracted data. All data were extracted at steps of 1.12 degrees to coincide with the average model grid-size. These bottom panels also show SISAL $\delta^{18}$O (circles for low-elevation sites, < 1,000 masl; triangles for high-elevation sites, > 1,000 masl).

**Review Figure 2:** Modern, Mid-Holocene (MH) and Last Glacial Maximum (LGM) transects for Oceania and SE Asia. Maps at the top of each panel show the simulated $\delta^{18}$Op from ECHAM5-wiso. All transects show absolute $\delta^{18}$O values. In the maps, filled circles are SISAL $\delta^{18}$O averages for entities that cover both the MH and the modern reference period. Filled squares are SISAL entities that do not have a corresponding modern. Bottom plots of each panel show the simulated data extracted for each transect: black circles and whiskers are mean $\pm 2$ standard deviation of the data extracted along longitudinal sections in between the two great circle lines shown in dashed grey lines in the top maps. The red line is the median of the extracted data. All data were extracted at steps of 1.12 degrees to coincide with the average model grid-size. These bottom panels also show SISAL $\delta^{18}$O (circles for low-elevation sites, < 1,000 masl; triangles for high-elevation sites, > 1,000 masl).

**Review Figure 3:** Speleothem samples for the period 1958-2013 converted to their drip-water equivalent using the fractionation factors from Grossman and Ku (1986; black dots) and Tremaine et al. (2011; red dots). We used simulated mean annual
temperature (MAT) for the years when samples are available for the conversion. Vertical lines indicate the offset thresholds for 0.1 and 0.3 with the former corresponding to the average isotope uncertainty in the SISAL database. Maximum offset occurs at low MAT and is 0.86.
* * *
[Figure]

[Figure]

**Review Figure 1:**

**Modern**  **MH**  **LGM**

[Figure]

ECHAM5-wiso w$\delta^{18}$O mean ±1 standard deviation
ECHAM5-wiso w$\delta^{18}$O median
SISAL $\delta^{18}$O (<1000 masl)
SISAL $\delta^{18}$O (>1000 masl)

SISAL / simulated weighted $\delta^{18}$O (averages; ‰ V-SMOW)

**Fig. 1.**

**Review Figure 2:**

**Modern**    **MH**    **LGM**

Fig. 2.

**Review Figure 3:**

Legend:
- (black) $\delta^{18}O_{dripw} = \delta^{18}O_{carbonate} - (18.34 \times 10^3 \times T^{-1} - 31.954)$; Grossman and Ku (1986)
- (red) $\delta^{18}O_{dripw} = \delta^{18}O_{carbonate} - (16.1 \times 10^3 \times T^{-1} - 24.6$; Tremaine et al . (2011)

Y-axis: $\delta^{18}O$ [V-SMOW]

X-axis: Mean Annual Temperature [deg C]

Offset = 0.30 ‰

Offset = 0.10 ‰

**Fig. 3.**

---

## Author Response (AR1)

**RE: Submission of revised manuscript cp-2019-25**

Dear Prof. Zhengtang Guo,

Please, find our point by point reply to the reviewers' comments below along with the manuscript with tracked changes.

We thank the reviewers for their comments, which we believe helped strengthen this study. In addition to their suggestions, we have done the following changes to the manuscript:

- We have incorporated the available aragonite speleothems into our analyses. A detailed explanation about how we have converted them to their drip-water equivalent is in lines 210-234 (section 2.3). Because of this change, we have revised all figures and tables.

- We have removed the Multivariate Analyses from the Supplementary Material and instead added one new figure (Figure S1) and two new tables (Table S1 and S2), which complement Figures S5 and S6.

- We have corrected for typos and revised sentences that were not clear.

All the changes done to the manuscript can be found after our point-by-point answers to the reviewers, with the changes are shown in red.

We hope that the revised manuscript satisfies both the reviewers and the yourself.

Many thanks for taking your time to deal with our manuscript. Please, do not hesitate to contact me with further questions or requests.

Best regards,

Laia Comas Bru

Post-doctoral researcher
University of Reading, UK

Please note that the reviewer's comments are in black and our answers in blue.

**Reply to anonymous Referee #1**

**1 Major comment: what is the added value of looking at spatial patterns for past climates compared to looking at them for present-day?**

The authors argue that it is useful for model evaluation to look at spatial patterns of absolute δ 18O for past climates (LGM, MH) rather than just looking at anomaly maps, in contrast with many previous studies. This allows to have more sites for modeldata comparison. However, what is the added value of looking at spatial patterns for past climates compared to looking at them for present-day? For present-day, spatial patterns would be the same to first order. At present, there are so many more sites available directly samplig precipitation (GNIP), so why bother with speleothem records for past climates?

Figures 7 and 8 show the spatial patterns of observed and simulated δ 18O for LGM and MH. The sub-figure a representing North and South America are common to both figures, and they actually show very similar patterns. The same figure for present day would also show very similar patterns. This is because δ18O, temperature or precipitation changes between LGM, MH and present-day are much smaller than the magnitude of spatial variations along a transect covering such a wide range of latitudes. So these figures support my skepticism about the relevance of spatial patterns in absolute values for past climates. Figures 7b,c and 8b,c do not represent the same regions. But I'm sure that the maps for present-day would look very similar.

The corresponding text bears several slips of the pen and/or interpretation errors, that probably reflect that writing this sub-section was not confortable:

– l 408: "δ18O changes" should be replaced by "δ18O patterns": the authors write "changes" because this is really what is interesting to look at, but actually the figures do not show it.

– l 412: "underestimates changes in precipitation": again, we cannot see changes from present-day to MH on this figure.

What you want to plot depends on the science question. If the science question is what controls spatial patterns in absolute values, then it's better to focus on present day values; past climates do not provide much added value. But if the science question is what controls the changes at paleo-climatic time scales, then it is necessary to look at anomalies between 2 climatic states.

So I recommend to remove section 3.4, or replace it by an analysis of spatial patterns of anomalies, and to modify accordingly the abstract, protocol and conclusions.

We have removed the old Figures 7 and 8 and substitute them for two new figures showing the spatial patterns across time-periods (i.e., modern, MH and LGM) across Europe and Asia. These new figures show that the gradients change over time due to the large ice-sheets during the LGM and the different insolation patterns at different latitudes during the MH. We use these figures to illustrate how the model is not able to simulate these gradients in some cases.

We have also decided to focus only on isotope data and have therefore removed the pollen-based reconstructions completely.

Please see the revised section 3.4 in lines 447-481.

**{2 Detailed comments**

**L157-158: I don't understand what this means. Where is the control simulations included in the LGM-MH difference?**

We have clarified this in the text with the following sentence (see lines 161-165):

*"We also calculated the anomaly between the LGM and MH (LGM-MH), taking account of the difference between their control simulations in the following way: $(lgm_{PI}-lgm) – (6ka_{PI}-6ka).$"*

**L209: remove "non-equilibrium of"**

We have done this (see lines 220-221)

**L216: add a dot.**

We have done this

**L223: remove "with data ... baseline".**

We have clarified this sentence by writing (see lines 239-241):

*"Data-model comparisons are generally made by comparing (1) anomalies between a control period and a palaeoclimate simulation with (2) data anomalies with respect to a modern baseline."*

**L234-235: already said, remove.**

We agree with the reviewer that this has already been mentioned in L156-158 and the text has been deleted.

**246: define pchip**

We have added the definition of pchip: "piecewise cubic hermite interpolation" in the text (see line 260)

**L295: clarify the rationale. Why can't there be a sampling bias in temperate regions towards the PMIP periods?**

The previous sentence already suggested that the deviations could result from a sampling bias. However, we have rephrased this sentence to clarify this point (see lines 316-321)):

*"These deviations could arise from sampling biases but it is unlikely that such biases would lead to differences between the tropics and temperate regions. Differences between curves constructed for both tropical and temperate regions (Fig. 2 c) suggest that, at least for the last 130 ka, deviations from expected stalagmite growth in the extra-tropics correspond to variability on glacial/interglacial scales."*

**L296: clarify this sentence. What does "even at a global level" mean?**

*We have deleted this sentence and replaced it by (see lines 323-326):*

*"Thus, the speleothem data indicate similar climatic sensitivity, even at a global level, to that demonstrated for sub-continental and regional scales by earlier authors, despite their use of much smaller numbers and far less precise age data than in the SISAL dataset."*

**L300-302: can you explain briefly why higher latitude speleothems are more depleted than OIPC and low latitude speleothems are more enriched?**

This comment is not relevant any more. The scatterplots in figure 3 were updated after incorporating aragonite speleothems in this study. The text describing the revised figure is in line 333.

**L305: "cave specific factors" cannot explain why you have such systematic differences common to wide regions.**

We refer to our answer above.

**L317-319: should the reader conclude that ECHAM underestimates the interannual variability? If so, please state this clearly. Has such a bias already been described in a previous paper, for ECHAM or for other models? Explain briefly what could be the reason for this underestimate.**

We have modified the manuscript to incorporate information on why ECHAM is underestimating d18O variability (see lines 358-363):

*"Our results are consistent with the general tendency of climate models to underestimate the sensitivity of extreme precipitation to temperature variability or trends (Flato et al., 2014). ECHAM5 is known to underestimate inter-annual variability in regions where precipitation is dominantly convective (i.e., the tropics), as well as in summer in extra-tropical regions (e.g., in southern Europe) because convective precipitation operates on small spatial scales and has a large random component, even for a given large-scale atmospheric state (Eden et al., 2012)."*

**L321: move "processes" before "within"**

We have revised this sentence to read (see line 365):

*"… reflecting the impact of karst and in-cave processes that effectively act as a low-pass filter…"*

**L325-328: this has already been said just above.**

We have now deleted the duplicated part of that sentence (see lines 369-372).

**L359: replace "anomalies" by "MH values"?**

We have made the suggested change in the text. See line 404.

**L422: remove "utilising"**

We have rephrased this sentence to (see line 483-484)

*"Our analyses illustrate a number of possible approaches for using speleothem isotopic data for model evaluation."*

**L440-447: this issue was not previously discussed. Add some quantification, or a map, showing what error we would make if we use only the fractionation factor for calcite?**

While the impact of using one of another fractionation equation is minimal (i.e. smaller than the measurement uncertainty) for sites with MAT > 27.3C, the added uncertainty is noticeable at sites with lower MAT. Changes of MAT across time periods could exacerbate these differences for individual sites thus making the calculated entity-based anomalies inaccurate. As requested by the reviewer, we have now added new supplementary figure (Fig. S1) to show that using the appropriate correction according to the speleothem's mineralogy is important.

**L473: remove "on a global basis". Or clarify what you mean. Even at a specific cave, if the speleothem acts as a low pass filter, time scales shorter than "quasidecadal" cannot be studied.**

We have modified that last section of this paragraph to clarify this point (see lines 536-538):

*The low variability shown by the SISAL records – most likely from the low-pass filter effectively applied to the speleothem record by the karst system – precludes the use of this database for global studies focused on time scales shorter than quasi-decadal."*

In addition, we have also added the following text in the results section 3.2 to clarify this point (see lines 372-375):

*"This result indicates that global data-model comparisons using speleothem records should focus on quasi-decadal or longer timescales. However, the temporal smoothing caused by karst processes varies from site to site; where transmission from the surface to the cave can be shown to be rapid, individual speleothems may preserve annual or even sub-annual signals."*

**L475-476: clarify. Do you mean that the model underestimates δ18O changes?**

Yes, the model underestimates the amplitude of $\delta^{18}O$ changes as recorded in the speleothems. For details we refer the reviewer to our answers on their comment on L317-319. We have rephrased this sentence to (see lines 540-543):

*"Using the traditional anomaly approach to data-model comparisons, there is consistency between the sign of observed and simulated changes in both the MH and the LGM exists. However, the ECHAM5-wiso model underestimates the changes in $\delta_{18}O$ between time periods compared to the speleothems records (i.e., the amplitude of modelled $\delta_{18}O$ changes is lower)."*

**L476-477: clarify.**

We agree that this sentence is too broad and general and have decided to delete it.

**L489: "constraining structural error on the model side": what do you mean? There are plenty of sources of errors in the model: errors on forcings, missing processes in parameterization package, tunable parameters, coarse resolution... Which one are you refering to? "true uncertainties": beware that errors are not the same as uncertainties. Anyway, in this paragraph, I suggest to focus only on uncertainties on the observation side, because this is what is useful to evaluate models. The question of quantifying uncertainties in models is set differently and is beyond the scope of this paper.**

We have rephrased this sentence to (see lines 506-515):

*"Mismatches between simulations and observations can reflect the issues with the experimental design, problems with the model or uncertainties in the observations (Harrison et al., 2015). The failure to include changes in atmospheric dust loading, for example, has been put forward as an explanation of data-model mismatches in both the MH and the LGM (e.g., Hopcroft et al., 2015; Messori et al., 2019). Missing processes and feedbacks, such as climate-induced vegetation or land-surface changes, could also contribute to mismatches (e.g., Yoshimori et al., 2009; Swann et al., 2014). Uncertainties caused by the specific structure of the model or assigned model parameter values could also contribute to data-model mismatches (Qian et al., 2016). Ultimately, there needs to be an assessment of the contribution of all of these factors to data-model mismatches, but here we have only focused on potential uncertainties associated with the speleothem data."*
* * *
**Reply to Anonymous Referee #2**

**Minor issues**

**Are the control runs for MH and LGM different? Probably I don't catch the points. In my understanding, they should be the same which is the base for probing the climatic significance of the difference between the MH and LGM simulation experiments.**

As suggested by reviewer 1, we have clarified how we have taken into account the difference in control runs for the MH and LGM simulations with the following sentence (see lines 161-165):

*"We also calculated the anomaly between the LGM and MH (LGM-MH), taking account of the difference between their control simulations in the following way: $(lgm_{PI}-lgm) – (6ka_{PI}-6ka)$."*

**The simulation results of the MH seem to be better than that of the LGM. Could they explain more on this? For example, they use the protocol of PIMP3 for the LGM modelling, and their SST forcing is based on the results of a full transient experiment. More clarification on why they take such steps will make this manuscript more convinced.**

Unfortunately, we cannot say that the MH simulation results fit better to the speleothem data compared to the LGM simulation due to the limited number of speleothem records available during the LGM.
* * *
**Reply to Anonymous Referee #3**

**I don't see very well what is the advancement made by this study as compared to the traditional approach for comparing the speleothem records with models. Maybe the authors should stress more why their approaches are better and what new can been discovered by their approaches that cannot be done by traditional approach.**

Data-model comparisons using speleothem data are comparatively new and have tended to focus on validation of new versions of isotope-enabled models. These comparisons have often overlooked important characteristics of, and/or uncertainties associated with, the speleothem records (see discussion section). There is no agreed protocol for using speleothem data for model evaluation. The purpose of our paper is to identify issues that could affect data-model comparisons, drawing on the new SISAL database that has been explicitly constructed to facilitate such comparisons and the expertise of the speleothem experts who constructed this database. Thus, we are not claiming that our approach is different from or better than a "traditional" approach – we are simply making it clear how speleothem data should and could be used. We have clarified the purpose of the paper by amplifying our description in L107-111 as follows:

*"In this paper, we examine a number of issues that need to be addressed in order to use speleothem data, specifically data from the SISAL database, for model evaluation in the palaeoclimate context and make recommendations about robust approaches that should be used for model evaluation in CMIP6-PMIP4. We focus particularly on interpretation issues that could be overlooked in using 110 speleothem records and we show the strengths and limitations of different comparison techniques."*

In addition, we have also expanded the conclusion section (see lines 582-594) to better clarify the purpose of this paper. Here we refer to the answer below.

**It is not very clear to me what is the final goal of the data-model comparison and what can be improved or learned after all the analyses. If the comparison is good, can we assume that the temperature and precipitation simulated by the model are correct and what is the uncertainty? What might be the reasons for the similarities and differences between model results and speleothem data? Can the results help to improve the model and/or experiment design and how?**

*As explained in the introduction (lines 41-53), model evaluation using palaeoclimate data provides an out-of-sample test of model performance and is one component of the Palaeoclimate Modelling Intercomparison Project. Such evaluations help to provide confidence in the projections of future climates. Speleothems are a relatively new source of information for such evaluations and the purpose of our paper is to provide a robust framework to make such evaluations. We do not want to distract from this goal by discussing the generic purposes of data-model comparison in the Introduction to the paper, but we have added a concluding paragraph discussing what can be learnt from such data-model comparisons as follows (L582-594):*

"Comparisons with speleothem data can be seen as a complement to model evaluation using other types of palaeoenvironmental data and palaeoclimatic reconstructions (see e.g. MARGO Project Members, 2009; Harrison et al., 2014). They can be considered particularly useful because they provide insights into how well state-of-the-art models reproduce the hydrological cycle and atmospheric circulation patterns. The ability to reproduce past observations provides additional confidence in the ability of climate models to simulate large climate changes, such as those expected by the end of the 21st century (Braconnot et al., 2012; Schmidt et al., 2014). However, mismatches between model simulations and palaeo-observations are also useful because they can help to pinpoint issues that may need to be addressed in developing improved models or in better experimental protocols (Kageyama et al., 2018), providing that these mismatches do not arise because of misunderstanding or misinterpretation of the observations themselves. By providing a protocol for using speleothem data for data-model comparisons that accounts for uncertainties in the observations, we anticipate that at least such causes of data-model mismatches will be minimized."

**The major uncertainties and biases of the ECHAM5-wiso model in simulating present day and past climates and the experiment design of the MH and LGM simulations, the reliability of the SST and sea ice simulated by the CCSM3 and their potential influence on the data-model comparison should be discussed.**

We use outputs from the ECHAM-wiso model in order to illustrate potential approaches to data-model comparison. Our goal here is not to provide an in-depth evaluation of the quality of these simulations. The performance of the ECHAM-wiso model under modern day conditions has been extensively analysed (see e.g. Werner et al., 2011; Wackerbarth et al., 2012) and the MH and LGM simulations have also been published and discussed (Wackerbarth et al., 2012; Werner et al., 2018). In order to make it clear that our use of the model is illustrative, we have revised the final section of the introduction to read:

*"We use an updated version of the SISAL database (SISALv1b: Atsawawaranunt et al., 2019) and simulations made with the ECHAM5-wiso isotope-enabled atmospheric circulation model (Werner et al., 2011) to explore the various issues in making data-model comparisons. The goal is not to evaluate the ECHAM5-wiso simulations but rather to use them to illustrate generic issues in data-model comparison with speleothem isotopic data."*

**The simulations for MH and LGM are only 12 and 22 years. Are they long enough to allow the climate at different speleothem location reaching equilibrium? What is the initial state of these simulations? What might be the influence of using fixed ocean condition?**

We explain in the methods section (lines 133-164) that these simulations are atmosphere-only simulations forced with sea-surface temperatures and sea-ice cover from a pre-existing transient simulation. Thus, there is no spin-up necessary and the issue of equilibrium is irrelevant. If the purpose of this paper were to use the model simulations to explain speleothem records, then the lack of ocean coupling would mean that the simulations would be unsuitable for evaluating the degree to which long-term (multi-decadal) variability in the speleothem isotope record reflected internal unforced variability. But as our purpose in using the experiments is illustrative, then the short length of the simulations is not important. We hope that the modification to the introduction mentioned above will help clarify the purpose of this paper.

[revised manuscript text omitted]